# Assessing effects of nature-based and other municipal adaptation measures on insured heavy rain damages

Vylon Ooms<sup>1,2</sup>, Thijs Endendijk<sup>1</sup>, Jeroen. C.J.H. Aerts<sup>1,3</sup>, W. J. Wouter Botzen<sup>1</sup>, Peter J. Robinson<sup>1</sup>

5

- 1. Institute for Environmental Studies (IVM) VU Amsterdam, Amsterdam, the Netherlands.
- 2. Dutch Association of Insurers, The Hague, the Netherlands.
- 3. Deltares Institute, Delft, The Netherlands.

Correspondence to: V. Ooms (v.ooms@vu.nl)

10 Key words:

**Key words:** Risk reduction – Adaptation – Nature-based – Insurance – Flooding – Rainfall

#### **Abstract**

Intense short duration rainfall events are expected to increase in severity and frequency due to climate change. Densely populated urban areas are vulnerable to these events, resulting in high losses. Implementing nature-based (e.g. green streets, rain gardens and green roofs) and other municipal adaptation measures (e.g. water storage facilities) can be a way to mitigate these damages. Little is known about the effectiveness of these measures combined in a municipality. This study assesses municipal climate adaptation measures being taken by the municipality of Amsterdam. Unique claims data of almost all Dutch insurers is used to understand the impact of these climate adaptation interventions. We study one neighborhood in Amsterdam which has been renovated using climate adaptation measures, including nature-based solutions. We implement a quasi-experimental difference-in-Differences (DiD) analysis that compares insured rainfall damages in the area to a similar neighboring area that was not renovated with climate adaptation measures. We find a negative significant relation between climate adaptation measures and insured damage when comparing the area where measures were taken to the similar area were measures were not taken, i.e. damage is reduced by climate adaptation measures by €1375-€5648 per rain day in the treatment area. Furthermore, precipitation per day is positively and significantly associated with insured damage. We suggest that nature-based and other adaptation measures can be installed by local governments and stimulated by insurers and banks to increase climate resilience in urban areas.

#### 1. Introduction

Densely built cities are vulnerable to intense short duration rainfall events, i.e. *cloudbursts* (Rosenzweig et al., 2019), which can result in pluvial flooding and high damage to buildings and infrastructure. For example, on the 2nd of July in Copenhagen a single cloudburst of extreme precipitation caused over €800 mln of damage (The City of Copenhagen, 2012). The rainfall event in Southern Germany in June 2024 reached €2-3bn of insured losses (MOODY's, 2024). Due to climate change, cloudbursts are likely to increase in frequency and severity (IPCC, 2022).

A wide range of resilience and additional flood adaptation measures are needed to cope with cloudbursts (Rosenzweig et al., 2018; Busker et al., 2021). Pluvial flood resilience in urban areas is often created by Flood Damage Mitigation (FDM) measures (e.g. water storage, drainage systems, etc.) taken by the (local) government. Furthermore, governments play a key role in enhancing resilience to flood damage, for example by investing in structural protection measures, such as dikes (Filatova, 2014). The traditional approach is engineering through building drainage systems, levees and dams. According to Sörensen et al. (2016), additional strategies are needed to enhance flood resilience such as adopting "blue-green infrastructure", like green roofs, rain gardens and porous pavements. These blue-green infrastructure can be used to retain (storm)water and therefore reduce flood risks (Sörensen et al., 2019).

There is also a role for households and businesses in flood damage risk reduction. For instance, they can implement emergency FDM measures (e.g. placing sandbags which act as a barrier to flood water and elevating personal possessions) and take structural FDM measures (e.g. making walls water-resistant and strengthening their buildings' foundation) (Endendijk et al., 2023). Moreover, insurance may be purchased to cover damages in cases where these measures fail. However, it has been shown that individuals, communities and businesses often underinvest in protection against low-probability, high-consequence flood events (Meyer & Kunreuther, 2017). Therefore, governments can undertake interventions to stimulate flood preparedness by households and businesses through awareness campaigns (Osberghaus & Hinrichs, 2020). Such awareness campaigns may focus on educating households about flood risk and potential coping strategies.

The goal of this study is to understand the impact of nature-based and other adaptation measures measures on insured damages caused by cloudbursts. The innovation of our study is threefold. Firstly, we examine the impact of municipal climate adaptation measures on insured damages empirically. A wide body of literature has assessed flood damage using mainly flood damage modelling methods (Merz et al., 2013; Spekkers et al., 2014; Van Ootegem et al., 2015). Traditional flood damage models focus on simulating flood depths of riverine flooding and estimating damage based on exposure information, such as building classes and their vulnerability (Merz et al., 2010; Sörensen & Mobini, 2017). However, multiple studies have shown that flood depth and building class information cannot fully explain flood damage, since it requires an extensive dataset which is often not available (Wagenaar et al., 2017; Merz et al., 2010). Moreover, few studies have studied pluvial flood risk modelling (Van Ootegem et al., 2015; Porter et al., 2023), which is the hazard focus of our study. Even fewer studies have investigated the effect of FDM measures on reducing damage caused by pluvial flooding (Löwe et al., 2017)<sup>1</sup>. Modelling studies focus on situations that are modelled, and therefore not observed in real life. Empirical studies, that include real damage observations, are needed to better understand the effectiveness of FDM measures. That is, empirical studies are more suitable for drawing conclusions from actual conditions, compared to conclusions derived from modelling studies that are typically based on assumed conditions.

The second novelty of this paper is that we use actual insurance damage data to identify causal effects of FDM measures. A small but expanding body of literature has focused on assessing the effectiveness of FDM measures on a household level using surveys as empirical methods (Endendijk et al., 2023; Kreibich et al., 2015; Poussin et al., 2015; Thieken et al., 2005). For example, Endendijk et al. (2023) found that household FDM measures reduced damage due to flooding by about 30% for

<sup>1</sup> One exception is Löwe et al. (2017), which examined the effect of 9 scenarios of urban development and 32 combinations of FDM measures on flood damages. They find that the effectiveness of the measures depends on climate and urban development. That is, these measures are interlinked, and the effectiveness can change through variations in climate, suggesting that a strategy with different measures through time is preferable to one-off investments.

buildings and 40% for home contents using survey data. Other studies show that FDM measures on a building level have substantial effects in limiting flood damage (Kreibich et al., 2015; Poussin et al., 2015; Thieken et al., 2005). In this research, we do not only focus on adaptation measures of individuals (e.g. green roofs), but also on spatial, neighbourhood level adaptation measures of the municipality. With survey data one can typically only identify correlational effects. In this study, we aim to identify causal effects with a quasi-experiment using real damage data from insurers. The Difference-in-Differences (DiD) method allows us to identify the causal effect of FDM measures (Angrist & Pischke, 2008). Also, in surveys it is possible that damages are misreported, whereas in this study we examine observed damages registered by insurance company professionals. The use of a DiD-method is an innovative addition to the existing literature on climate adaptation (Osberghaus & Hinrichs, 2020). In this study, we illustrate how a DiD-method can work in the climate adaptation field.

The third innovation of this study is that we assess the effectiveness of a broad range of policy interventions, including naturebased solutions. In the literature, most studies examine the effect of a single FDM measure or policy intervention in isolation (Osberghaus & Hinrichs, 2020; Sörensen & Emilsson, 2019). More comprehensive approaches may be needed for substantial flood risk reduction (Busker et al., 2022; Osberghaus & Hinrichs, 2020). Osberghaus & Hinrichs (2020) is, to the best of our knowledge, the only study that adopts a quasi-experimental design to assess the effectiveness of an FDM measure. They use a 90 DiD-design to measure the impact of a large-scale flood risk awareness campaign from 2009 to 2017 on flood damage (as well as households' adaptation behaviour and insurance penetration) in Germany. They do not find a significant effect of the awareness campaign on flood damages. Another study on a single FDM measure is done by Sörensen & Emilsson (2019), who assessed the effectiveness of a stormwater system retrofitted through climate adaptation using insurance claims data. They find that long term trends show less flood damage in the area with these adaptation measures compared to similar neighborhoods. 95 There are studies that focus on the impact of single measures like retrofitting an old stormwater system (Sörensen & Emilsson, 2019), blue-green roofs (Busker et al., 2021) or awareness campaigns (Osberghaus & Hinrichs, 2020). This paper studies a broader range of interventions such as awareness campaigns by adding climate adaptation measures to the study as well. In reality, a wide array of measures is needed to reduce damage resulting from cloudbursts (Busker et al., 2022). We lack understanding of the impact of a broad range of FDM measures on insured damages.

The remainder of this paper is structured as follows. Section 2 gives an overview of the methodology. Section 3 gives the results that are discussed in Section 4. The conclusion follows in section 5.

#### 2. Methodology

115

80

#### 2.1 Case study description

In this study we use insurance claims data to understand the impact of municipal adaptation interventions on pluvial flood damages in Amsterdam. We focus on parts of the city where such interventions have been implemented over time. We use data on the timing of specific interventions provided by the program *Amsterdam Weerproof* (Amsterdam Weatherproof), which aims to make the city more climate resilient. In this program, various structural measures have been implemented, like retrofitting municipality owned buildings into greener properties, creating more green areas, improving water storage locations, and sewer renewal. Moreover, another focus of the organization is to provide extreme weather information to raise awareness of flood risk of citizens through online and in-person information provision (Amsterdam Weerproof, 2024).

Amsterdam Weerproof executed projects in various neighbourhoods. We compare two adjacent areas of the neighbourhood *Rivierenbuurt* with different postal codes (PC). In PC 1078, *Scheldebuurt* (treatment area), municipal adaptation measures were executed from 2018 until 2022. We compare this neighborhood to PC 1079, *Rijnbuurt* (control area), where no measures were taken. Detailed descriptions of the Rivierenbuurt neighbourhood are found in Appendix 1. Table 1 describes the climate adaptation measures that were taken in the Scheldebuurt.

Table 1. Adopted nature-based and other adaptation measures in the treatment area (Amsterdam Weerproof, 2025).

| Type of measure                         | Explanation of measure                                                                                                                                                                                                                                                                     |
|-----------------------------------------|--------------------------------------------------------------------------------------------------------------------------------------------------------------------------------------------------------------------------------------------------------------------------------------------|
| Municipal spatial                       | nature-based and other adaptation measures                                                                                                                                                                                                                                                 |
| Renewal of the sewer system             | Renewal of the sewer system in the Scheldebuurt                                                                                                                                                                                                                                            |
| Extra green areas                       | Creation of green areas next to roads                                                                                                                                                                                                                                                      |
| Water storage squares                   | Installation of water storage capacity at a square (Europaplein) and under tram lanes                                                                                                                                                                                                      |
| Allocated spaces for water to flow into | Installation of water storage areas in streets and the creation of larger green spaces around trees for water to flow into.                                                                                                                                                                |
| Household and bu                        | siness level nature-based measures                                                                                                                                                                                                                                                         |
| Rain proofing advice                    | Free garden advice from Amsterdam Rainproof coaches on how to make your property more rainproof (e.g. replacing tiles for greenery in gardens and green roofing). This was incentivized by a municipal subsidy, for instance for replacing tiles of 15 euro per m <sup>2</sup> .           |
| Additional green spaces                 | The addition of small gardens in front of privately owned property, incentivized by the municipality. Inhabitants of Amsterdam can ask the municipality for a garden in front of their house. Then, the municipality will remove the tiles and build a small garden in front of the house. |

#### 120 2.2 Data

#### 2.2.1 Pluvial flood insurance claims data and nature-based and other adaptation measures

For this study we use claims data of rain damage of households from the Dutch Association of Insurers. The Dutch Association of Insurers registers claims of households filed by insurance companies that are members of the association. Since rain damage is covered by default in property and contents insurance products (Dutch Association of Insurers, 2025), we expect that the vast majority of the claims are accepted. More than 95% of the Dutch insurers market is member of the Dutch Association of Insurers (Dutch Association of Insurers, 2024). Furthermore, more than 95% of households with a contents and/or property insurance in the Netherlands are insured against rain damage (Dutch Association of Insurers, 2016). Therefore, almost all pluvial flood damages of households in the studied neighbourhoods are reflected in the insurance claims. We use aggregated data on postcode 4-level (PC 4)<sup>2</sup> for the municipality of Amsterdam (2007-2024). In the Netherlands, PC 4 refers to a neighbourhood or a part of a district within a municipality. The damage data ranges from January 1<sup>st</sup> 2007 until March 15<sup>th</sup> 2024. The rain damage claims consist of time (day), amount (damage in euros) and location (at PC 4-level).

The treatment variable is the observed time from when nature-based and other adaptation measures were implemented. From November 1st 2018 onwards the municipality of Amsterdam implemented nature-based and other climate adaptation measures to reduce damage in the treatment area with PC 1078 (Amsterdam Weerproof, 2025).

<sup>&</sup>lt;sup>2</sup> Due to privacy restrictions on the claims data it is not possible to look at the damages on address level.

Both models we use showcase the analysis without the intervention period, to make for a cleaner analysis of the comparison before and after the implementation of the measures<sup>3</sup>. This choice has been made to avoid potential bias from including the rollout period, when the policy's effect was only partial.

**Table 2**. Dependent variable, treatment variables and their descriptive statistics over the final sample period (excluding the intervention period).

| Variable                                          | Variable description                                                                                                                                                                                                                                        | Data source                         | Mean<br>deviation if<br>in parenthese |                       |
|---------------------------------------------------|-------------------------------------------------------------------------------------------------------------------------------------------------------------------------------------------------------------------------------------------------------------|-------------------------------------|---------------------------------------|-----------------------|
|                                                   |                                                                                                                                                                                                                                                             |                                     | From 2007                             | From<br>2016          |
| Dependent var                                     | iables                                                                                                                                                                                                                                                      |                                     |                                       |                       |
| Insured rain damage                               | Amount of insured damage per day in the Rivierenbuurt caused by rain claimed at an insurer operating in the Netherlands in euros.                                                                                                                           | Dutch<br>Association of<br>Insurers | €198.410<br>(€2,070.290)              | €242.315<br>(€30.000) |
| Treatment vari                                    | iables                                                                                                                                                                                                                                                      |                                     |                                       |                       |
| Treatment:<br>Municipal<br>adaptation<br>measures | Binary variable. 1 = When the observation is part of the treatment area where climate adaptive interventions have been taken. 0 = when the observation is in the control area, where no adaptation intervention took place during the study period.         | Amsterdam<br>Weerproof              | 0.500                                 | 0.500                 |
| Post                                              | Binary variable. 1 = Observation after end of intervention period of Februari 1 <sup>st</sup> 2022, when municipal adaptation measures in the treatment area have been taken. 0 = observations before intervention period of November 1 <sup>st</sup> 2018. | Amsterdam<br>Weerproof              | 0.210                                 | 0.596                 |

## 2.2.2 Rain data and socio-demographic characteristics

Control variables are added to check for neighborhood specific effects when establishing the relationship between the adaptation measures and the amount of damage. Two categories of variables are controlled for. Precipitation data is added on PC4 level over the period damage data is available from January 1st 2007 until the March 15th 2024. The nearest weather station of the Royal Netherlands Meteorological Institute (KNMI) is located at Schiphol airport, which is approximately 10 km from the Rivierenbuurt. Two types of data are derived from the weather station: data on amount of precipitation per day and data on maximum precipitation per hour. Both are included, because moderate rain over a long period within a day can cause damage as well as torrential rain in a short moment. The observations of the damage data are on the day on which the claim is filed. The claim can be filed on the same day as the event that caused the damage. However, people can also file claims one or two days later. Therefore, for both rain control variables we use one- and two-day lags.

Additionally, data on socio-demographic characteristics of the Rivierenbuurt (e.g. average house price and average size of households) is used to control for neighborhood specific effects. This data is derived from public data of Statistics Netherlands (CBS), which is only available on a yearly basis from January 1<sup>st</sup> 2016 until December 31<sup>st</sup> 2023. Therefore, we interpolate between the years to create daily neighborhood characteristic data and extrapolate in the period January 1<sup>st</sup> 2024 until March

\_

<sup>&</sup>lt;sup>3</sup> The results with the intervention period included are used as a robustness test and can be observed in Appendix 4. An analysis with only the significant variables with the intervention period is included in Appendix 6.

15<sup>th</sup> 2024 by assuming linear trends. In Appendix 5 we show tables on the distribution of insured rain damage, a detailed description of insured rain damage data and the distribution of rain data.

**Table 3.** Control variables and their descriptive statistics over the final sample period (excluding the intervention period).

| Variable                   | Variable description                                                                                                                 | Data<br>source | Mean and stand<br>non-binary | dard deviation if       |
|----------------------------|--------------------------------------------------------------------------------------------------------------------------------------|----------------|------------------------------|-------------------------|
| Rain Data                  |                                                                                                                                      |                | From 2007                    | From 2016               |
| Sum of rain per day        | Sum of rain in 0.1 mm at the weather station around Schiphol airport (the nearest station is approximately 10 km from Rivierenbuurt) | KNMI           | 23.157<br>(48.208)           | 24.003<br>(50.030)      |
| Max sum of rain in an hour | Max sum of rain in an hour at Schiphol airport in 0.1 mm                                                                             | KNMI           | 8.793 (18.220)               | 8.904 (17.820)          |
| Area characteris           | stics (per day from 2016)                                                                                                            |                |                              |                         |
| Population density         | The amount of people per km <sup>2</sup>                                                                                             | Statistics N   | letherlands                  | 13,907.740<br>(805.956) |
| Building charac            | teristics(per day from 2016)                                                                                                         |                |                              |                         |
| Value property             | Average price per real estate asset based on the Valuation of Immovable Property Act (WOZ) (€x1000).".                               | Statistics N   | letherlands                  | 475.500<br>(133.814)    |

### 2.3 Difference-in-difference method

155

170

In this study, we use a DiD two-way fixed effects model to estimate the impact of municipal adaptation measures on rainfall damage in Amsterdam. This method compares a situation before and after an intervention period. We compare two adjacent areas within the Rivierenbuurt neighborhood: one where FDM measures have been implemented (Scheldebuurt) and another where no adaptation interventions have been implemented (Rijnbuurt). The DiD-approach allows us to compare changes in outcomes over time between these areas, while controlling for unobserved factors and broader trends (Card & Krueger, 1993;
 Wooldridge, 2014). By leveraging insurance claims data, we can isolate the causal impact of these measures under the assumption that both areas would have followed similar trends in the absence of interventions. We test this assumption in the next section.

We expand upon a traditional DiD by employing a two-way fixed effects (TWFE-) model (Callaway & Sant'Anna, 2021). Using fixed effects in a DiD gives a more robust causal estimate. This approach controls for time-invariant unobserved differences between neighborhoods, such as historical infrastructure and socioeconomic factors, as well as time-specific shocks, like extreme weather events. By accounting for both unit (neighborhood) and time (month) fixed effects, the TWFE-model ensures that our estimated treatment effect reflects the impact of adaptation measures rather than underlying trends or external influences. This strengthens the causal interpretation of the DiD-analysis. We estimate the following TWFE-model:

$$Y_{it} = \beta_0 + \beta_1 treatment_i \times post_t + \beta_2 X'_{it} + \delta_i + \theta_t + \varepsilon_{it}$$

The outcome variable Y<sub>it</sub> represents daily insured damage claims in euros. Moreover, we expect that no rain damage occurs with slight rain (<2 mm/h). Therefore, we look at cases of moderate, or higher rain (>2 mm/h) in classification (Met Office UK, 2012). Excess rainfall can accumulate on the surface and may cause damage to buildings. Therefore, we only include

damage observations linked to days when this threshold is exceeded, along with a two-day lag period to account for potential delays in damage claims reporting. The average treatment effect is given by  $\beta_1$ , which captures the average impact of the policy interventions in the treated area in the TWFE specification (Callaway & Sant'Anna, 2021). We control for time-invariant neighborhood differences using unit (postcode 4-level) fixed effects ( $\delta_i$ ). Time-specific neighborhood-level shocks are controlled for through fixed effects for each month ( $\theta_t$ ). The coefficient vector of other control variables is represented by  $\beta_2$ , and the error term is given by  $\epsilon_{it}$ .

#### 2.4 Common trend assumption

180

The central assumption for a DiD-analysis is the common trend assumption, which states that, in the absence of the treatment, the treatment and control groups would have followed a similar trend (in our case of insured damages) over time (Angrist & Pischke, 2008). This assumption allows for isolating the treatment effect from any other factors that may influence damage from rainfall. If both neighbourhoods were on different damage trajectories before the policy interventions, differences in their post-interventions outcomes could be attributed to these pre-treatment differences. Additionally, it is assumed that no significant changes in group composition occur over time. Data from Statistics Netherlands indicates that there were no shocks to the demographic composition of the neighbourhoods during the study period, supporting this assumption. Moreover, key demographics are controlled for in our regression model.

The placebo test can be performed to check for the common trend assumption (Eggers et al., 2021). The placebo test checks the common trend assumption by creating "fake" treatment groups before and after the interventions. We select a different treatment timeframe and observe whether the effects are significant as well. If no effect is found in any of the placebo groups, it supports that the found treatment effect can be attributed to the treatment rather than pre-existing trends. Angrist and Pischke (2008) used lag and lead values of treatment status to show that no significant effects occurred in the placebo periods. In Appendix 2, we apply placebo tests by using one- and two-month leads and lags for the treatment variable. These placebo treatment variables resulted in non-significant outcomes, reinforcing the validity of the common trend assumption for causal inference.

#### 3. Results

The results are shown in Table 4 for two models. The first model showcases the results of the dataset starting from 2007 until 2024 without control variables for area characteristics, which are unavailable for this entire time period. In this model we see that the DiD-indicator shows a significant (p < 0.05) reduction of insured damage in the treatment compared to the control group. This means that in the area where nature-based and other adaptation measures were adopted, insured damage in the treatment group is on average  $\{0.375$  per day lower for rain events exceeding 2 mm per hour as compared to the control group, after controlling for time- and unit fixed effects. The second model presents results using damage data starting from 2016, when area characteristics are available as control variables. The coefficient on the interaction term shows a significant (p < 0.05) reduction of damage in the treatment group, compared to the control group. The rain damage is, on average, lower by  $\{0.05\}$  reduction of damage in the treatment group, compared to the control group. The rain damage is, on average, lower by  $\{0.05\}$  reduction of 21,7% for rainy days with more than  $\{0.05\}$  of insured damage per year on average based on model 1 (from 2007).

\_

<sup>&</sup>lt;sup>4</sup> The damage reduction could be illustrated using an example by the following steps. Firstly, there are 35 rainy days on which severe damage (more than €2,500 of insured damage) occurred. Second, when we divide these rainy days by the 17.21 years in the dataset (January 1, 2007 – March 15, 2024), we obtain 2.03 rainy days with more than €2,500 of insured damage per year on average. Third, the total damage in the treatment area on 35 rainy days with more that €2,500 of damage is €221,771.20. Fourth, if we divide this amount by the 17.21 years in the dataset, we obtain €12,886.18 of damage per year on rainy days with more than €2500 of insured damage, on average. Next, according to model 1 (from 2007), insured damage in the treatment group is on average €1,375 per rainy day lower compared to the control group. There are 2.03 rainy days with more than €2,500 of insured damage per year. If we multiply the coefficient (€1,375) times 2.03 rainy days, we obtain €2,791,25 of damage reduction on these days. Lastly, when we divide €2,791.25 by the total €12,886.18 of damage per year on rainy days with more than €2,500 of insured damage, obtain 0,217. Here, we can observe a damage reduction of 21,7% for rainy days with more than €2,500 of insured damage per year on average based on model 1 (from 2007).

Furthermore, the variable for precipitation per day is positive and significant (p < 0.01) in model 1, indicating that an increase of 0.1 mm precipitation per day results in an increase of  $\in 13.75$  of rain damage on average per rain day based on model 1. Regarding the area- and building characteristics control variables, we see that none of the variables are significantly associated with insured damage. According to the adjusted R-squared, model 1 explains 16.7% of the variation in insured damage and model 2 explains 17.3% of the variation.

**Table 4.** Two-way fixed effects DiD regression on insured damage per day in case of maximum rain per hour exceeds 2mm per hour from 2007 and 2016 without observations in the intervention period.

|                                           | (1)       | (2)      |
|-------------------------------------------|-----------|----------|
|                                           | (1)       | (2)      |
| Variables                                 | Model 1   | Model 2  |
| D4 V 44 (D:D)                             | 1 275**   | £ (40**  |
| Post × treatment (DiD)                    | -1,375**  | -5,648** |
| G C : 1 (' 0.1                            | (558.201) | (2,512)  |
| Sum of rain per day (in 0.1 mm)           | 6.856***  | 7.100    |
|                                           | (2.308)   | (5.375)  |
| Sum of rain per day lag 1 (in 0.1 mm)     | -2.010    | -0.986   |
|                                           | (3.635)   | (9.271)  |
| Sum of rain per day lag 2 (in 0.1 mm)     | -0.083    | 0.624    |
| ,                                         | (4.274)   | (11.800) |
| Maximum rain in an hour (in 0.1 mm)       | -3.053    | -13.902  |
|                                           | (6.315)   | (16.893) |
| Maximum rain in an hour lag 1 (in 0.1 mm) | 10.883    | 14.205   |
|                                           | (9.674)   | (27.483) |
| Maximum rain in an hour lag 2 (in 0.1 mm) | -0.665    | 2.483    |
| ,                                         | (12.785)  | (37.700) |
| Population density (per km²)              | •         | -6.391   |
|                                           |           | (5.845)  |
| Value of property (in euros)              |           | 48.002   |
|                                           |           | (56.332) |
| Constant                                  | -61.365   | 66,643   |
|                                           | (285.967) | (98,878) |
| Observations                              | 1,416     | 536      |
| R-squared                                 | 0.259     | 0.271    |
| Adjusted R-squared                        | 0.167     | 0.173    |
| Standard errors in parentheses            |           |          |

Standard errors in parentheses

#### 4. Discussion and recommendations

#### 4.1 Discussion of findings in relation to the existing literature

Impact nature-based and other adaptation measures measures on rain damage (post × treatment): In both models we find a significant reduction of insured damages in the treatment group compared to the control group. The interaction result of model

<sup>\*\*\*</sup> p<0.01, \*\* p<0.05, \* p<0.1

1 is impacted by high damage observations in August 2010 in the control group compared to the treatment group. This also explains why the standard deviation is very high compared to the average of damage data (Table 2) and rain data (Table 3). The results of the impact of nature-based and other adaptation measures on damage are in line with some previous studies on physical adaptation measures. Sörensen & Emilsson (2019) present trends showing less damage in areas with adaptation measures compared to similar neighbourhoods. Also, the findings are in line with studies on the stated effectiveness of FDM measures: Endendijk et al. (2023), Kreibich et al. (2015), Poussin et al. (2015), and Thieken et al. (2005) all confirm the damage reductive capacity of flood risk reduction measures. The addition of this study is the DiD-design, which allows us to identify the causal effect of FDM measures. To our knowledge, the method is hardly seen in the climate adaptation field. We illustrate with this that this method can work. Future studies could adopt this method as well in different areas.

Rain control variables: Model 1 shows a significant result regarding precipitation per day. Contrastingly, the precipitation per day variable in model 2 is insignificant. Model 1 has 1416 observations and model 2 has 536 observations. The fact that model 2 has less than half the number of observations could be an explanation why no significant coefficient is found for the rain control variables in model 2. The literature findings on the relation between rain and damage vary. Previous literature on pluvial floods and damage show that flood depth (among other factors) cannot fully explain damage (Wagenaar et al., 2017; Merz et al., 2010). However, Sörensen et al. (2017) do find that rainfall intensity is one of the main determinants of flood damage. We further do not find a significant relation between damage and maximum rain per hour.

#### 4.2 Policy implications

We find that nature-based and other adaptation measures reduce rain damage. Local governments can use nature based and other adaptation measures (e.g. through green lanes, water storage facilities, green roofs, and greener gardens) as means to decrease rain damage in urban areas and increase livability and biodiversity in these areas (Skrydstrup et al., 2022). These nature-based measures often come with co-benefits like mental and physical benefits (Tzoulas et al., 2007), which can have a long-term impact on health by incentivizing people to exercise, for instance. Rain damage is the focus of this study. The measures the municipality applied could also limit impacts of other natural hazards, like drought (Ljubojević et al., 2025) and heat (Augusto et al., 2020). In this way, nature-based measures can limit long-term impacts of climate change in the area (Augusto et al., 2020). The benefits (in addition to the damage reducing potential of these measures) make these nature-based solutions attractive for designing climate resilient cities globally. The measures the city of Amsterdam implemented (e.g. water storage on city squares, green roofs) can be implemented in cities worldwide. The findings of this study can motivate national governments, building corporations, and project developers to construct buildings and infrastructure in a climate adaptive way. The quantification of avoided damage can also be useful for cost-benefit analyses. Measures like green roofs and rain gardens can be stimulated by governments using policy measures like subsidies. Lastly, the results of this study can motivate insurers to stimulate the uptake of climate adaptive measures of their customers. Insurers could stimulate these measures by providing flood risk information or giving premium discounts when customers take climate adaptive measures and may benefit from lower claims (Poussin et al., 2015; Mol et al., 2020).

#### 4.3 Limitations and research implications

In this study we use insurance damage data. Most studies using insurance data use data of a single insurer (Cheng et al., 2012) or only a few insurers (Sörensen et al., 2019). A strength of this study is the use of high-resolution insurance data covering more than 95% of the Dutch insurance market (Dutch Association of Insurers, 2024). However, the data contains only household claims, and we here neglect insurance claims of businesses and uninsured damages. It would be of value to analyze uninsured damages (e.g., public infrastructure) and claims of businesses as well. Insured damage of households is only a part of total damage of extreme rain, but can still give valuable insights into the effectiveness of FDM measures. The fact that only two full years (2022-2024) had passed since the end intervention period could be a limitation. However, we do find significant

<sup>&</sup>lt;sup>5</sup> In an additional analysis, we omitted the month August 2010, with the large damages in the treatment group and the control group. This month is an outlier and seemed to impact the interaction result and the coefficient. We see some changes in the results: the interaction coefficient is -1432, compared to the -1375 in the model with August 2010 included, and the relation is significant on a higher level (p < 0.01).

effects already. Moreover, torrential rain can be a local event, whereas we used rain data measured at the nearest weather station of which data may deviate from the real rainfall at the case study locations. This difference in data granularity between local insured damages and rainfall may weaken statistical significance between these two variables and means that the rainfall data may lack precision. Additionally, it would be insightful if future research could be conducted on social vulnerability (e.g. financial situation), since that could influence insurance uptake. Furthermore, this study shows the impact of all adaptation measures combined. Because of privacy regulations, it was not possible to localize claims on a more detailed level than PC 4-level. This makes it difficult to attach effects of a single measure to single damage claims. It would be valuable to understand how much separate measures contribute to damage reduction. This would give information on which measures policymakers could prioritize. In a future study, it might be of value to understand the impact of these measures separately.

#### 5. Conclusion

In this study, we show the impact of various nature-based and other adaptation measures on insured rain damage. We add novel insights to the literature by using actual insurance damage data to identify causal effects of a broad range of adaptation measures. Our results show a robust significant reduction in damage caused by the adoption of climate adaptation measures in the city of Amsterdam. The effect of nature-based and other climate adaptation measures on rain damage suggests that governments, private investors, banks and insurers can stimulate and implement these measures to cope with increasing rain damage. Local governments can incentivize the uptake of these measures among their citizens through information provision and subsidization. Private investors can invest in climate adaptive real estate to finance durable, resilient real estate and infrastructure that can withstand heavy rain damage. Banks can stimulate climate adaptation by including adaptation measures for resilient houses in loans (e.g. climate adaptive mortgage products). Insurers can stimulate climate adaptation measures through information provision, premium discounts and climate adaptive retrofitting (building back better) after damage. Improving the understanding of the impact of climate adaptation measures is important to increase societal climate resilience. Cloudbursts can increase in severity and frequency, potentially causing more floods in urban areas. The implementation of nature-based and other adaptation measures is important to prevent urban floods and reduce damage in urban areas globally.

Data availability The insurance claims data of the Dutch Association of Insurers is not available publicly due to privacy or ethical restrictions.

Competing interests. V. Ooms has received funding from the Dutch Association of Insurers.

Author contribution. VO: formal analysis, data curation, writing (original draft preparation and review and editing), and methodology. TE: methodology and writing (specifically on the methodology section). JCJHA: supervision and writing (review and editing). WJWB: conceptualization, supervision and writing (review and editing). PJR: conceptualization, supervision and writing (review and editing).

*Financial support.* V. Ooms received funding from the Dutch Association of Insurers. W.J.W. Botzen received funding from the project NATURANCE (grant No 101060464) of the Horizon 2020 Framework Programme of the European Union. J.C.J.H. Aerts received funding from the ERC COASTMOVE project nr884442.

Acknowledgements: The authors would like to thank the Dutch Association of Insurers for making the data available.

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

## Appendix A. Description Rivierenbuurt.

In *Rivierenbuurt* we compare two parts of the same neighbourhood. One where measures have been executed, *Scheldebuurt*, and one where no measures have been taken, *Rijnbuurt*.

Table A1. Description Rivierenbuurt

| 4.635 people          | 14.580 people                                                          |
|-----------------------|------------------------------------------------------------------------|
|                       |                                                                        |
| JI                    | 110                                                                    |
| 990                   | 1625                                                                   |
| 685 addresses per km2 | 6106 addresses per km2                                                 |
| UR 650.000            | EUR 541.000                                                            |
| 80% between 1925-1950 | >80% between 1925-1950                                                 |
| 8,3% apartments       | 98,9% apartments                                                       |
| 9%                    | 20%                                                                    |
| 8                     | 585 addresses per km2 UR 650.000 80% between 1925-1950 8,3% apartments |

#### Appendix B. Placebo tests.

435

The goal of this placebo test is to identify whether groups were experiencing similar trends before the treatment. This can be done by creating 'fake' treatments that indicate treatment before it actually occurred (Angrist & Pischke, 2009). These placebo treatments should have no effect if the common trend assumption holds. If they do show significant effects, this suggests a violation of the assumption, as it indicates that treated and control groups were already on diverging paths prior to the interventions.

We apply placebo tests by using one- and two-month leads and lags for the treatment variable. The lead and lagged placebo treatments do not show any significant outcomes, which provides evidence in favour of the common trend assumption.

**Table B1.** Placebo test 2007 with lags of one month and two months.

| Variables                               | (1)                 |
|-----------------------------------------|---------------------|
| 1.treatment                             | <del>-</del>        |
| 0bL30.placebo                           | 0                   |
| 0020000                                 | (0)                 |
| 1oL30.placebo                           | 0                   |
|                                         | (0)                 |
| 0b.treatment#0bL30.placebo              | 0                   |
|                                         | (0)                 |
| 0b.treatment#1oL30.placebo              | 0                   |
|                                         | (0)                 |
| 1o.treatment#0bL30.placebo              | 0                   |
|                                         | (0)                 |
| 1.treatment#1L30.placebo                | 220.6               |
|                                         | (2,906)             |
| 0bL60.placebo                           | 0                   |
|                                         | (0)                 |
| 1oL60.placebo                           | 0                   |
| 01                                      | (0)                 |
| 0b.treatment#0bL60.placebo              | 0                   |
| 01                                      | (0)                 |
| 0b.treatment#1oL60.placebo              | 0                   |
| 1 4 4 4 4 4 4 4 4 4 4 4 4 4 4 4 4 4 4 4 | (0)                 |
| 1o.treatment#0bL60.placebo              | 0                   |
| 1 444#11 (011 -                         | (0)                 |
| 1.treatment#1L60.placebo                | 95.85               |
| Raindepth                               | (2,879)<br>6.921*** |
| Kanidepui                               | (2.352)             |
| lag1_Raindepth                          | -1.991              |
| iag i_ixamucpui                         | (3.687)             |
| lag2 Raindepth                          | -0.0509             |
| ing2_italindoptii                       | (4.332)             |
| Rainhourmax                             | -3.115              |
| Tummourmus                              | (6.411)             |
| lag1 Rainhourmax                        | 10.92               |
| 1451411110 011110/1                     | 10.72               |

|                  | (9.802) |
|------------------|---------|
| lag2 Rainhourmax | -0.687  |
|                  | (12.98) |
| Constant         | -230.6  |
|                  | (286.1) |
| Observations     | 1,388   |
| R-squared        | 0.255   |

**Table B2.** Placebo test 2007 with leads of one month and two months.

| Variables                                                                                                                                                                                                                                                                                                                                                                                                                                                                                                                                                                                                                                                                                                                                                                                                                                                                                                                                                                                                                                                                                                                                                                                                                                                                                                                                                                                                                                                                                                                                                                                                                                                                                                                                                                                                                                                                                                                                                                                                                                                                                                                    |                                |          |
|------------------------------------------------------------------------------------------------------------------------------------------------------------------------------------------------------------------------------------------------------------------------------------------------------------------------------------------------------------------------------------------------------------------------------------------------------------------------------------------------------------------------------------------------------------------------------------------------------------------------------------------------------------------------------------------------------------------------------------------------------------------------------------------------------------------------------------------------------------------------------------------------------------------------------------------------------------------------------------------------------------------------------------------------------------------------------------------------------------------------------------------------------------------------------------------------------------------------------------------------------------------------------------------------------------------------------------------------------------------------------------------------------------------------------------------------------------------------------------------------------------------------------------------------------------------------------------------------------------------------------------------------------------------------------------------------------------------------------------------------------------------------------------------------------------------------------------------------------------------------------------------------------------------------------------------------------------------------------------------------------------------------------------------------------------------------------------------------------------------------------|--------------------------------|----------|
| Do.treatment                                                                                                                                                                                                                                                                                                                                                                                                                                                                                                                                                                                                                                                                                                                                                                                                                                                                                                                                                                                                                                                                                                                                                                                                                                                                                                                                                                                                                                                                                                                                                                                                                                                                                                                                                                                                                                                                                                                                                                                                                                                                                                                 |                                | (1)      |
| ObF30.placebo         0           1oF30.placebo         0           0b.treatment#0bF30.placebo         0           0b.Rivierenbuurt#1oF30.placebo         0           0b.Rivierenbuurt#1oF30.placebo         0           1o.treatment#0bF30.placebo         0           1.treatment#1F30.placebo         103.7           0bF60.placebo         0           1oF60.placebo         0           0b.treatment#0bF60.placebo         0           0b.treatment#1oF60.placebo         0           0b.treatment#1oF60.placebo         0           1c.treatment#1F60.placebo         0           1c.treatment#1F60.placebo <td>Variables</td> <td></td>                                                                                                                                                                                                                                                                                                                                                                                                                                                                                                                                                                                                                                                                                                                                                                                                                                                                                                       | Variables                      |          |
| ObF30.placebo         0           1oF30.placebo         0           0b.treatment#0bF30.placebo         0           0b.Rivierenbuurt#1oF30.placebo         0           0b.Rivierenbuurt#1oF30.placebo         0           1o.treatment#0bF30.placebo         0           1.treatment#1F30.placebo         103.7           0bF60.placebo         0           1oF60.placebo         0           0b.treatment#0bF60.placebo         0           0b.treatment#1oF60.placebo         0           0b.treatment#1oF60.placebo         0           1c.treatment#1F60.placebo         0           1c.treatment#1F60.placebo <td></td> <td></td>                                                                                                                                                                                                                                                                                                                                                                                                                                                                                                                                                                                                                                                                                                                                                                                                                                                                                                                |                                |          |
| 10F30.placebo                                                                                                                                                                                                                                                                                                                                                                                                                                                                                                                                                                                                                                                                                                                                                                                                                                                                                                                                                                                                                                                                                                                                                                                                                                                                                                                                                                                                                                                                                                                                                                                                                                                                                                                                                                                                                                                                                                                                                                                                                                                                                                                | 1o.treatment                   | <u> </u> |
| 10F30.placebo                                                                                                                                                                                                                                                                                                                                                                                                                                                                                                                                                                                                                                                                                                                                                                                                                                                                                                                                                                                                                                                                                                                                                                                                                                                                                                                                                                                                                                                                                                                                                                                                                                                                                                                                                                                                                                                                                                                                                                                                                                                                                                                |                                |          |
| 10F30.placebo         0           0b.treatment#0bF30.placebo         0           0b.Rivierenbuurt#10F30.placebo         0           10.treatment#0bF30.placebo         0           1.treatment#1F30.placebo         0           0bF60.placebo         0           0bF60.placebo         0           0b.treatment#0bF60.placebo         0           0b.treatment#0bF60.placebo         0           0b.treatment#1oF60.placebo         0           10.treatment#1bF60.placebo         0           1.treatment#1F60.placebo         0           1.treatment#1F60.placebo         280.3           1.gaindepth         7.118***           1.gaindepth         -2.162           3.835         1ag2 Raindepth         -0.173           4.533         3.415           6.491)         6.491)                                                                                                                                                                                                                                                                                                                                                                                                                                                                                                                                                                                                                                                                                                                                                                                                                                                                                                                                                                                                                                                                                                                                                                                                                                                                                                                                          | 0bF30.placebo                  |          |
| (0)   (0)   (0)   (0)   (0)   (0)   (0)   (0)   (0)   (0)   (0)   (0)   (0)   (0)   (0)   (0)   (0)   (0)   (0)   (0)   (0)   (0)   (0)   (0)   (0)   (0)   (0)   (0)   (0)   (0)   (0)   (0)   (0)   (0)   (0)   (0)   (0)   (0)   (0)   (0)   (0)   (0)   (0)   (0)   (0)   (0)   (0)   (0)   (0)   (0)   (0)   (0)   (0)   (0)   (0)   (0)   (0)   (0)   (0)   (0)   (0)   (0)   (0)   (0)   (0)   (0)   (0)   (0)   (0)   (0)   (0)   (0)   (0)   (0)   (0)   (0)   (0)   (0)   (0)   (0)   (0)   (0)   (0)   (0)   (0)   (0)   (0)   (0)   (0)   (0)   (0)   (0)   (0)   (0)   (0)   (0)   (0)   (0)   (0)   (0)   (0)   (0)   (0)   (0)   (0)   (0)   (0)   (0)   (0)   (0)   (0)   (0)   (0)   (0)   (0)   (0)   (0)   (0)   (0)   (0)   (0)   (0)   (0)   (0)   (0)   (0)   (0)   (0)   (0)   (0)   (0)   (0)   (0)   (0)   (0)   (0)   (0)   (0)   (0)   (0)   (0)   (0)   (0)   (0)   (0)   (0)   (0)   (0)   (0)   (0)   (0)   (0)   (0)   (0)   (0)   (0)   (0)   (0)   (0)   (0)   (0)   (0)   (0)   (0)   (0)   (0)   (0)   (0)   (0)   (0)   (0)   (0)   (0)   (0)   (0)   (0)   (0)   (0)   (0)   (0)   (0)   (0)   (0)   (0)   (0)   (0)   (0)   (0)   (0)   (0)   (0)   (0)   (0)   (0)   (0)   (0)   (0)   (0)   (0)   (0)   (0)   (0)   (0)   (0)   (0)   (0)   (0)   (0)   (0)   (0)   (0)   (0)   (0)   (0)   (0)   (0)   (0)   (0)   (0)   (0)   (0)   (0)   (0)   (0)   (0)   (0)   (0)   (0)   (0)   (0)   (0)   (0)   (0)   (0)   (0)   (0)   (0)   (0)   (0)   (0)   (0)   (0)   (0)   (0)   (0)   (0)   (0)   (0)   (0)   (0)   (0)   (0)   (0)   (0)   (0)   (0)   (0)   (0)   (0)   (0)   (0)   (0)   (0)   (0)   (0)   (0)   (0)   (0)   (0)   (0)   (0)   (0)   (0)   (0)   (0)   (0)   (0)   (0)   (0)   (0)   (0)   (0)   (0)   (0)   (0)   (0)   (0)   (0)   (0)   (0)   (0)   (0)   (0)   (0)   (0)   (0)   (0)   (0)   (0)   (0)   (0)   (0)   (0)   (0)   (0)   (0)   (0)   (0)   (0)   (0)   (0)   (0)   (0)   (0)   (0)   (0)   (0)   (0)   (0)   (0)   (0)   (0)   (0)   (0)   (0)   (0)   (0)   (0)   (0)   (0)   (0)   (0)   (0)   (0)   (0)   (0)   (0)   (0)   (0)   (0)   (0)  |                                |          |
| Ob.treatment#0bF30.placebo         0           Ob.Rivierenbuurt#1oF30.placebo         0           Io.treatment#0bF30.placebo         0           I.treatment#1F30.placebo         103.7           (1,965)         (1,965)           ObF60.placebo         0           IoF60.placebo         0           (0)         (0)           Ob.treatment#0bF60.placebo         0           (0)         (0)           Io.treatment#1oF60.placebo         0           (0)         (0)           I.treatment#1F60.placebo         0           (1,867)         Raindepth           7.118***         (2,411)           lag1 Raindepth         -2.162           (3,835)         (4,533)           Rainhourmax         -3.415           (6,491)         (6,491)                                                                                                                                                                                                                                                                                                                                                                                                                                                                                                                                                                                                                                                                                                                                                                                                                                                                                                                                                                                                                                                                                                                                                                                                                                                                                                                                                                               | 1oF30.placebo                  |          |
| (0)   (0)   (0)   (0)   (0)   (0)   (0)   (0)   (0)   (0)   (0)   (0)   (0)   (0)   (0)   (0)   (0)   (0)   (0)   (0)   (0)   (1)   (1)   (1)   (1)   (1)   (1)   (1)   (1)   (1)   (1)   (1)   (1)   (1)   (1)   (1)   (1)   (1)   (1)   (1)   (1)   (1)   (1)   (1)   (1)   (1)   (1)   (1)   (1)   (1)   (1)   (1)   (1)   (1)   (1)   (1)   (1)   (1)   (1)   (1)   (1)   (1)   (1)   (1)   (1)   (1)   (1)   (1)   (1)   (1)   (1)   (1)   (1)   (1)   (1)   (1)   (1)   (1)   (1)   (1)   (1)   (1)   (1)   (1)   (1)   (1)   (1)   (1)   (1)   (1)   (1)   (1)   (1)   (1)   (1)   (1)   (1)   (1)   (1)   (1)   (1)   (1)   (1)   (1)   (1)   (1)   (1)   (1)   (1)   (1)   (1)   (1)   (1)   (1)   (1)   (1)   (1)   (1)   (1)   (1)   (1)   (1)   (1)   (1)   (1)   (1)   (1)   (1)   (1)   (1)   (1)   (1)   (1)   (1)   (1)   (1)   (1)   (1)   (1)   (1)   (1)   (1)   (1)   (1)   (1)   (1)   (1)   (1)   (1)   (1)   (1)   (1)   (1)   (1)   (1)   (1)   (1)   (1)   (1)   (1)   (1)   (1)   (1)   (1)   (1)   (1)   (1)   (1)   (1)   (1)   (1)   (1)   (1)   (1)   (1)   (1)   (1)   (1)   (1)   (1)   (1)   (1)   (1)   (1)   (1)   (1)   (1)   (1)   (1)   (1)   (1)   (1)   (1)   (1)   (1)   (1)   (1)   (1)   (1)   (1)   (1)   (1)   (1)   (1)   (1)   (1)   (1)   (1)   (1)   (1)   (1)   (1)   (1)   (1)   (1)   (1)   (1)   (1)   (1)   (1)   (1)   (1)   (1)   (1)   (1)   (1)   (1)   (1)   (1)   (1)   (1)   (1)   (1)   (1)   (1)   (1)   (1)   (1)   (1)   (1)   (1)   (1)   (1)   (1)   (1)   (1)   (1)   (1)   (1)   (1)   (1)   (1)   (1)   (1)   (1)   (1)   (1)   (1)   (1)   (1)   (1)   (1)   (1)   (1)   (1)   (1)   (1)   (1)   (1)   (1)   (1)   (1)   (1)   (1)   (1)   (1)   (1)   (1)   (1)   (1)   (1)   (1)   (1)   (1)   (1)   (1)   (1)   (1)   (1)   (1)   (1)   (1)   (1)   (1)   (1)   (1)   (1)   (1)   (1)   (1)   (1)   (1)   (1)   (1)   (1)   (1)   (1)   (1)   (1)   (1)   (1)   (1)   (1)   (1)   (1)   (1)   (1)   (1)   (1)   (1)   (1)   (1)   (1)   (1)   (1)   (1)   (1)   (1)   (1)   (1)   (1)   (1)   (1)   (1)   (1)   (1)   (1)   (1)   (1)   (1)   (1)  |                                |          |
| 0b.Rivierenbuurt#1oF30.placebo         0           10.treatment#0bF30.placebo         0           1.treatment#1F30.placebo         103.7           0bF60.placebo         0           10F60.placebo         0           0b.treatment#0bF60.placebo         0           0b.treatment#1oF60.placebo         0           0b.treatment#1bF60.placebo         0           1.treatment#1F60.placebo         0           1.treatment#1F60.placebo         0           1.treatment#1F60.placebo         280.3           1.treatment#1F60.placebo         280.3           1.treatment#1F60.placebo         -2.162           (2.411)         1ag1 Raindepth         -2.162           (3.835)         1ag2 Raindepth         -0.173           (4.533)         Rainhourmax         -3.415           (6.491)         (6.491)                                                                                                                                                                                                                                                                                                                                                                                                                                                                                                                                                                                                                                                                                                                                                                                                                                                                                                                                                                                                                                                                                                                                                                                                                                                                                                               | 0b.treatment#0bF30.placebo     |          |
| 1.treatment#0bF30.placebo                                                                                                                                                                                                                                                                                                                                                                                                                                                                                                                                                                                                                                                                                                                                                                                                                                                                                                                                                                                                                                                                                                                                                                                                                                                                                                                                                                                                                                                                                                                                                                                                                                                                                                                                                                                                                                                                                                                                                                                                                                                                                                    |                                | 3 /      |
| 10.treatment#0bF30.placebo       0         1.treatment#1F30.placebo       103.7         0bF60.placebo       0         10F60.placebo       0         0b.treatment#0bF60.placebo       0         0b.treatment#10F60.placebo       0         0       (0)         10.treatment#0bF60.placebo       0         10.treatment#0bF60.placebo       0         1.treatment#1F60.placebo       280.3         Raindepth       7.118***         1ag1 Raindepth       -2.162         1ag2 Raindepth       -0.173         2ag2 Raindepth       -0.173         Rainhourmax       -3.415         (6.491)       (6.491)                                                                                                                                                                                                                                                                                                                                                                                                                                                                                                                                                                                                                                                                                                                                                                                                                                                                                                                                                                                                                                                                                                                                                                                                                                                                                                                                                                                                                                                                                                                         | 0b.Rivierenbuurt#1oF30.placebo |          |
| (0)   1.treatment#1F30.placebo   103.7   (1,965)   0bF60.placebo   0   (0)   10F60.placebo   0   (0)   10F60.placebo   0   (0)   (0)   (0)   (0)   (0)   (0)   (0)   (0)   (0)   (0)   (0)   (0)   (0)   (0)   (0)   (0)   (0)   (0)   (0)   (0)   (0)   (0)   (0)   (1.treatment#10F60.placebo   0   (0)   (0)   (1.treatment#10F60.placebo   280.3   (1,867)   (1,867)   (1,867)   (2.411)   (2.411)   (2.411)   (2.411)   (2.411)   (2.411)   (2.411)   (2.411)   (2.411)   (2.411)   (2.411)   (2.411)   (2.411)   (2.411)   (2.411)   (2.411)   (2.411)   (2.411)   (2.411)   (2.411)   (2.411)   (2.411)   (2.411)   (2.411)   (2.411)   (2.411)   (2.411)   (2.411)   (2.411)   (2.411)   (2.411)   (2.411)   (2.411)   (2.411)   (2.411)   (2.411)   (2.411)   (2.411)   (2.411)   (2.411)   (2.411)   (2.411)   (2.411)   (2.411)   (2.411)   (2.411)   (2.411)   (2.411)   (2.411)   (2.411)   (2.411)   (2.411)   (2.411)   (2.411)   (2.411)   (2.411)   (2.411)   (2.411)   (2.411)   (2.411)   (2.411)   (2.411)   (2.411)   (2.411)   (2.411)   (2.411)   (2.411)   (2.411)   (2.411)   (2.411)   (2.411)   (2.411)   (2.411)   (2.411)   (2.411)   (2.411)   (2.411)   (2.411)   (2.411)   (2.411)   (2.411)   (2.411)   (2.411)   (2.411)   (2.411)   (2.411)   (2.411)   (2.411)   (2.411)   (2.411)   (2.411)   (2.411)   (2.411)   (2.411)   (2.411)   (2.411)   (2.411)   (2.411)   (2.411)   (2.411)   (2.411)   (2.411)   (2.411)   (2.411)   (2.411)   (2.411)   (2.411)   (2.411)   (2.411)   (2.411)   (2.411)   (2.411)   (2.411)   (2.411)   (2.411)   (2.411)   (2.411)   (2.411)   (2.411)   (2.411)   (2.411)   (2.411)   (2.411)   (2.411)   (2.411)   (2.411)   (2.411)   (2.411)   (2.411)   (2.411)   (2.411)   (2.411)   (2.411)   (2.411)   (2.411)   (2.411)   (2.411)   (2.411)   (2.411)   (2.411)   (2.411)   (2.411)   (2.411)   (2.411)   (2.411)   (2.411)   (2.411)   (2.411)   (2.411)   (2.411)   (2.411)   (2.411)   (2.411)   (2.411)   (2.411)   (2.411)   (2.411)   (2.411)   (2.411)   (2.411)   (2.411)   (2.411)   (2.411)   (2.411)   (2.411)   (2.411)   (2.411)   ( |                                | 3 /      |
| 1.treatment#1F30.placebo       103.7         0bF60.placebo       0         10F60.placebo       0         0b.treatment#0bF60.placebo       0         0b.treatment#1oF60.placebo       0         0       00         1o.treatment#0bF60.placebo       0         1.treatment#1F60.placebo       280.3         1.treatment#1F60.placebo       280.3         Raindepth       7.118***         (2.411)       1ag1 Raindepth       -2.162         (3.835)       1ag2 Raindepth       -0.173         (4.533)       Rainhourmax       -3.415         (6.491)       (6.491)                                                                                                                                                                                                                                                                                                                                                                                                                                                                                                                                                                                                                                                                                                                                                                                                                                                                                                                                                                                                                                                                                                                                                                                                                                                                                                                                                                                                                                                                                                                                                             | 1o.treatment#0bF30.placebo     |          |
| 0bF60.placebo       0         1oF60.placebo       0         0b.treatment#0bF60.placebo       0         0b.treatment#1oF60.placebo       0         0c       0         1o.treatment#0bF60.placebo       0         1.treatment#1F60.placebo       280.3         (1,867)       7.118***         Raindepth       7.118***         (2.411)       1ag1 Raindepth         -2.162       (3.835)         lag2 Raindepth       -0.173         (4.533)       Rainhourmax         Rainhourmax       -3.415         (6.491)                                                                                                                                                                                                                                                                                                                                                                                                                                                                                                                                                                                                                                                                                                                                                                                                                                                                                                                                                                                                                                                                                                                                                                                                                                                                                                                                                                                                                                                                                                                                                                                                                |                                |          |
| 0bF60.placebo       0         1oF60.placebo       0         0b.treatment#0bF60.placebo       0         0b.treatment#1oF60.placebo       0         0c       0         1o.treatment#0bF60.placebo       0         1.treatment#1F60.placebo       280.3         (1,867)       7.118***         (2.411)       1ag1 Raindepth       -2.162         (3.835)       1ag2 Raindepth       -0.173         (4.533)       (4.533)         Rainhourmax       -3.415         (6.491)                                                                                                                                                                                                                                                                                                                                                                                                                                                                                                                                                                                                                                                                                                                                                                                                                                                                                                                                                                                                                                                                                                                                                                                                                                                                                                                                                                                                                                                                                                                                                                                                                                                       | 1.treatment#1F30.placebo       |          |
| (0)   10F60.placebo                                                                                                                                                                                                                                                                                                                                                                                                                                                                                                                                                                                                                                                                                                                                                                                                                                                                                                                                                                                                                                                                                                                                                                                                                                                                                                                                                                                                                                                                                                                                                                                                                                                                                                                                                                                                                                                                                                                                                                                                                                                                                                          |                                | · · · ·  |
| 10F60.placebo       0         0b.treatment#0bF60.placebo       0         0b.treatment#1oF60.placebo       0         10.treatment#0bF60.placebo       0         1.treatment#1F60.placebo       280.3         Raindepth       7.118***         (2.411)       1         lag1 Raindepth       -2.162         (3.835)       1         lag2 Raindepth       -0.173         (4.533)       Rainhourmax         Rainhourmax       -3.415         (6.491)                                                                                                                                                                                                                                                                                                                                                                                                                                                                                                                                                                                                                                                                                                                                                                                                                                                                                                                                                                                                                                                                                                                                                                                                                                                                                                                                                                                                                                                                                                                                                                                                                                                                              | 0bF60.placebo                  |          |
| 0b.treatment#0bF60.placebo       0         0b.treatment#1oF60.placebo       0         10.treatment#0bF60.placebo       0         1.treatment#1F60.placebo       280.3         Raindepth       7.118***         (2.411)       1ag1 Raindepth         1ag2 Raindepth       -2.162         (3.835)       -0.173         Rainhourmax       -3.415         (6.491)       (6.491)                                                                                                                                                                                                                                                                                                                                                                                                                                                                                                                                                                                                                                                                                                                                                                                                                                                                                                                                                                                                                                                                                                                                                                                                                                                                                                                                                                                                                                                                                                                                                                                                                                                                                                                                                  |                                |          |
| 0b.treatment#0bF60.placebo       0         0b.treatment#1oF60.placebo       0         1o.treatment#0bF60.placebo       0         1.treatment#1F60.placebo       280.3         (1,867)       7.118***         Raindepth       7.2.162         (3.835)       (3.835)         lag2 Raindepth       -0.173         (4.533)       (4.533)         Rainhourmax       -3.415         (6.491)                                                                                                                                                                                                                                                                                                                                                                                                                                                                                                                                                                                                                                                                                                                                                                                                                                                                                                                                                                                                                                                                                                                                                                                                                                                                                                                                                                                                                                                                                                                                                                                                                                                                                                                                        | 1oF60.placebo                  |          |
| (0)         0b.treatment#1oF60.placebo       0         1o.treatment#0bF60.placebo       0         1.treatment#1F60.placebo       280.3         Raindepth       7.118***         (2.411)       (2.411)         lag1 Raindepth       -2.162         (3.835)       (4.533)         Rainhourmax       -3.415         (6.491)                                                                                                                                                                                                                                                                                                                                                                                                                                                                                                                                                                                                                                                                                                                                                                                                                                                                                                                                                                                                                                                                                                                                                                                                                                                                                                                                                                                                                                                                                                                                                                                                                                                                                                                                                                                                     |                                |          |
| 0b.treatment#1oF60.placebo       0         1o.treatment#0bF60.placebo       0         1.treatment#1F60.placebo       280.3         Raindepth       7.118***         (2.411)       (2.411)         lag1 Raindepth       -2.162         (3.835)       (4.533)         Rainhourmax       -3.415         (6.491)                                                                                                                                                                                                                                                                                                                                                                                                                                                                                                                                                                                                                                                                                                                                                                                                                                                                                                                                                                                                                                                                                                                                                                                                                                                                                                                                                                                                                                                                                                                                                                                                                                                                                                                                                                                                                 | 0b.treatment#0bF60.placebo     |          |
| 10.treatment#0bF60.placebo       0         1.treatment#1F60.placebo       280.3         Raindepth       7.118***         (2.411)       (2.411)         lag1 Raindepth       -2.162         (3.835)       (4.533)         Rainhourmax       -3.415         (6.491)                                                                                                                                                                                                                                                                                                                                                                                                                                                                                                                                                                                                                                                                                                                                                                                                                                                                                                                                                                                                                                                                                                                                                                                                                                                                                                                                                                                                                                                                                                                                                                                                                                                                                                                                                                                                                                                            |                                | , ,      |
| 1o.treatment#0bF60.placebo       0         1.treatment#1F60.placebo       280.3         Raindepth       (1,867)         Raindepth       (2.411)         lag1 Raindepth       -2.162         (3.835)       (3.835)         lag2 Raindepth       -0.173         (4.533)       (4.533)         Rainhourmax       -3.415         (6.491)                                                                                                                                                                                                                                                                                                                                                                                                                                                                                                                                                                                                                                                                                                                                                                                                                                                                                                                                                                                                                                                                                                                                                                                                                                                                                                                                                                                                                                                                                                                                                                                                                                                                                                                                                                                         | 0b.treatment#1oF60.placebo     |          |
| 1.treatment#1F60.placebo       280.3         Raindepth       (1,867)         Raindepth       7.118***         (2.411)       (2.411)         lag1 Raindepth       -2.162         (3.835)       (3.835)         lag2 Raindepth       -0.173         (4.533)       (4.533)         Rainhourmax       -3.415         (6.491)                                                                                                                                                                                                                                                                                                                                                                                                                                                                                                                                                                                                                                                                                                                                                                                                                                                                                                                                                                                                                                                                                                                                                                                                                                                                                                                                                                                                                                                                                                                                                                                                                                                                                                                                                                                                     |                                |          |
| 1.treatment#1F60.placebo       280.3         Raindepth       (1,867)         Raindepth       (2.411)         lag1 Raindepth       -2.162         (3.835)       (4.533)         Rainhourmax       -3.415         (6.491)                                                                                                                                                                                                                                                                                                                                                                                                                                                                                                                                                                                                                                                                                                                                                                                                                                                                                                                                                                                                                                                                                                                                                                                                                                                                                                                                                                                                                                                                                                                                                                                                                                                                                                                                                                                                                                                                                                      | lo.treatment#0bF60.placebo     |          |
| Raindepth       7.118***         (2.411)       (2.411)         lag1 Raindepth       -2.162         (3.835)       (3.835)         lag2 Raindepth       -0.173         (4.533)       (4.533)         Rainhourmax       -3.415         (6.491)                                                                                                                                                                                                                                                                                                                                                                                                                                                                                                                                                                                                                                                                                                                                                                                                                                                                                                                                                                                                                                                                                                                                                                                                                                                                                                                                                                                                                                                                                                                                                                                                                                                                                                                                                                                                                                                                                  |                                |          |
| Raindepth       7.118***         (2.411)         lag1 Raindepth       -2.162         (3.835)         lag2 Raindepth       -0.173         (4.533)       (4.533)         Rainhourmax       -3.415         (6.491)                                                                                                                                                                                                                                                                                                                                                                                                                                                                                                                                                                                                                                                                                                                                                                                                                                                                                                                                                                                                                                                                                                                                                                                                                                                                                                                                                                                                                                                                                                                                                                                                                                                                                                                                                                                                                                                                                                              | 1.treatment#1F60.placebo       |          |
| lag1 Raindepth       -2.162         (3.835)       -0.173         lag2 Raindepth       -0.173         (4.533)       -3.415         (6.491)       -0.173                                                                                                                                                                                                                                                                                                                                                                                                                                                                                                                                                                                                                                                                                                                                                                                                                                                                                                                                                                                                                                                                                                                                                                                                                                                                                                                                                                                                                                                                                                                                                                                                                                                                                                                                                                                                                                                                                                                                                                       |                                |          |
| lag1 Raindepth       -2.162         (3.835)       -0.173         lag2 Raindepth       -0.173         (4.533)       -3.415         (6.491)       -0.173                                                                                                                                                                                                                                                                                                                                                                                                                                                                                                                                                                                                                                                                                                                                                                                                                                                                                                                                                                                                                                                                                                                                                                                                                                                                                                                                                                                                                                                                                                                                                                                                                                                                                                                                                                                                                                                                                                                                                                       | Raindepth                      |          |
| (3.835)         lag2 Raindepth       -0.173         (4.533)         Rainhourmax       -3.415         (6.491)                                                                                                                                                                                                                                                                                                                                                                                                                                                                                                                                                                                                                                                                                                                                                                                                                                                                                                                                                                                                                                                                                                                                                                                                                                                                                                                                                                                                                                                                                                                                                                                                                                                                                                                                                                                                                                                                                                                                                                                                                 |                                |          |
| lag2 Raindepth       -0.173         (4.533)       (4.533)         Rainhourmax       -3.415         (6.491)       (6.491)                                                                                                                                                                                                                                                                                                                                                                                                                                                                                                                                                                                                                                                                                                                                                                                                                                                                                                                                                                                                                                                                                                                                                                                                                                                                                                                                                                                                                                                                                                                                                                                                                                                                                                                                                                                                                                                                                                                                                                                                     | lag1_Raindepth                 |          |
| Rainhourmax -3.415 (6.491)                                                                                                                                                                                                                                                                                                                                                                                                                                                                                                                                                                                                                                                                                                                                                                                                                                                                                                                                                                                                                                                                                                                                                                                                                                                                                                                                                                                                                                                                                                                                                                                                                                                                                                                                                                                                                                                                                                                                                                                                                                                                                                   | 1.001.1                        | ` /      |
| Rainhourmax -3.415 (6.491)                                                                                                                                                                                                                                                                                                                                                                                                                                                                                                                                                                                                                                                                                                                                                                                                                                                                                                                                                                                                                                                                                                                                                                                                                                                                                                                                                                                                                                                                                                                                                                                                                                                                                                                                                                                                                                                                                                                                                                                                                                                                                                   | lag2 Raindepth                 |          |
| (6.491)                                                                                                                                                                                                                                                                                                                                                                                                                                                                                                                                                                                                                                                                                                                                                                                                                                                                                                                                                                                                                                                                                                                                                                                                                                                                                                                                                                                                                                                                                                                                                                                                                                                                                                                                                                                                                                                                                                                                                                                                                                                                                                                      |                                |          |
|                                                                                                                                                                                                                                                                                                                                                                                                                                                                                                                                                                                                                                                                                                                                                                                                                                                                                                                                                                                                                                                                                                                                                                                                                                                                                                                                                                                                                                                                                                                                                                                                                                                                                                                                                                                                                                                                                                                                                                                                                                                                                                                              | Rainhourmax                    |          |
| lag1_Rainhourmax 11.10                                                                                                                                                                                                                                                                                                                                                                                                                                                                                                                                                                                                                                                                                                                                                                                                                                                                                                                                                                                                                                                                                                                                                                                                                                                                                                                                                                                                                                                                                                                                                                                                                                                                                                                                                                                                                                                                                                                                                                                                                                                                                                       |                                | ` /      |
|                                                                                                                                                                                                                                                                                                                                                                                                                                                                                                                                                                                                                                                                                                                                                                                                                                                                                                                                                                                                                                                                                                                                                                                                                                                                                                                                                                                                                                                                                                                                                                                                                                                                                                                                                                                                                                                                                                                                                                                                                                                                                                                              | lag l Rainhourmax              | 11.10    |

|                              | (10.00) |
|------------------------------|---------|
| ag2_Rainhourmax              | -0.357  |
|                              | (13.11) |
| Constant                     | -236.1  |
|                              | (288.1) |
| bservations                  | 1,384   |
| R-squared                    | 0.255   |
| andard errors in parentheses |         |

Standard errors in parentheses \*\*\* p<0.01, \*\* p<0.05, \* p<0.1

#### 445 **Table B3.** Placebo test 2016 with lags of one month and two months.

|                              | (1)          |
|------------------------------|--------------|
| Variables                    |              |
|                              |              |
| 1.treatment                  | <del>-</del> |
|                              |              |
| 0bL30.placebo                | 0            |
|                              | (0)          |
| 1oL30.placebo                | 0            |
|                              | (0)          |
| 0b.treatment#0bL30.placebo   | 0            |
|                              | (0)          |
| 0b.treatment#1oL30.placebo   | 0            |
| •                            | (0)          |
| 1o.treatment#0bL30.placebo   | 0            |
|                              | (0)          |
| 1.treatment#1L30.placebo     | 1,015        |
|                              | (5,998)      |
| 0bL60.placebo                | 0            |
|                              | (0)          |
| 1oL60.placebo                | 0            |
|                              | (0)          |
| 0b.treatment#0bL60.placebo   | 0            |
| oon cannelly obboth acces    | (0)          |
| 0b.treatment#1oL60.placebo   | 0            |
| vo.treatment#10E00.ptace00   | (0)          |
| 1o.treatment#0bL60.placebo   | 0            |
| 10.treatment// 00E00.ptace00 | (0)          |
| 1.treatment#1L60.placebo     | 2,349        |
| 1.treatment// 1E00.place00   | (5,952)      |
| Raindepth                    | 7.159        |
| канасри                      | (5.656)      |
| lag1 Raindepth               | -1.352       |
| iagi_Rainuchii               | (9.680)      |
| lag2 Raindepth               | 0.545        |
| iagz Kaniucpin               |              |
|                              | (12.34)      |

| Rainhourmax                    | -13.73    |
|--------------------------------|-----------|
|                                | (17.64)   |
| lag1 Rainhourmax               | 15.63     |
|                                | (28.68)   |
| lag2 Rainhourmax               | 3.387     |
|                                | (39.49)   |
| populationdensity              | -4.207    |
|                                | (7.381)   |
| WOZwaardewoning                | 10.16     |
|                                | (59.88)   |
| Constant                       | 53,174    |
|                                | (126,043) |
| Observations                   | 504       |
| R-squared                      | 0.266     |
| Standard errors in parentheses |           |

Standard errors in parentheses
\*\*\* p

Figure C1. Median insured rain damage per year in treatment area and control area.

Note: The interventions began in November 2018 and was finished January 31st 2022. To be closest to this, we drew the lines of the start of the intervention in 2018 (in grey). The end of the intervention is drawn in 2021 (in black), since 11 out of 12 months of 2022 have taken place after the intervention period. Also, in the figure above there is no datapoint for 2024, because there are no observations of damages and rainfall exceeding 2 mm per hour in 2024 (until 15th of March).

480

470

## Appendix D. DiD regression with intervention period.

**Table D1.** Two-way fixed effects DiD regression on insured damage per day in case of maximum rain per hour exceeds 2mm per hour with observations in the intervention period.

|                                           | (3)        |      | (4)       |      |
|-------------------------------------------|------------|------|-----------|------|
| Variables                                 | Model      | 3    | Model     | 4    |
| variables                                 | (results   | from | (results  |      |
|                                           | 2007       | with | 2016      | with |
|                                           | interventi |      | intervent |      |
|                                           | period)    | OII  | period)   | 1011 |
|                                           | periou)    |      | periou)   |      |
| Post × treatment (DiD)                    | -646.963   | k    | -5,017**  | *    |
| Tost weamon (BIB)                         | (392.473)  |      | (1,863)   |      |
| Sum of rain per day (in 0.1 mm)           | 6.267***   | ,    | 5.405     |      |
|                                           | (1.949)    |      | (3.650)   |      |
| Sum of rain per day lag 1 (in 0.1 mm)     | -2.158     |      | -1.136    |      |
| ·                                         | (3.004)    |      | (5.774)   |      |
| Sum of rain per day lag 2 (in 0.1 mm)     | 0.434      |      | 2.537     |      |
|                                           | (3.423)    |      | (6.711)   |      |
| Maximum rain in an hour (in 0.1 mm)       | -2.679     |      | -8.205    |      |
| ·                                         | (5.075)    |      | (9.669)   |      |
| Maximum rain in an hour lag 1 (in 0.1 mm) | 10.618     |      | 11.500    |      |
|                                           | (7.733)    |      | (15.412)  |      |
| Maximum rain in an hour lag 2 (in 0.1 mm) | -2.997     |      | -8.074    |      |
|                                           | (9.572)    |      | (18.218)  |      |
| Population density (per km²)              |            |      | -7.325**  | *    |
|                                           |            |      | (2.827)   |      |
| Value of property (in euros)              |            |      | 24.671    |      |
|                                           |            |      | (27.787)  |      |
| Constant                                  | -62.000    |      | 91,656**  |      |
|                                           | (240.200)  |      | (40,384)  |      |
| Observations                              | 1,766      |      | 886       |      |
| R-squared                                 | 0.254      |      | 0.269     |      |
| Adjusted R-squared                        | 0.162      |      | 0.174     |      |
| Standard errors in parentheses            |            |      |           |      |

Standard errors in parentheses

<sup>\*\*\*</sup> p<0.01, \*\* p<0.05, \* p<0.1

## Appendix E. Distributions and descriptions of data

**Table E1**. Distribution insured rain damage data full dataset (from 2007).

|                             | 1%    | 5%          | 10%         | 25%         | 50%         | 75%           | 90%           | 95%           | 99%            | Largest          |
|-----------------------------|-------|-------------|-------------|-------------|-------------|---------------|---------------|---------------|----------------|------------------|
| All observation s (n=12568) | €0.00 | €0.00       | €0.00       | €0.00       | €0.00       | €0.00         | €169.00       | €1,000.<br>00 | €3,761.00      | €169,305.<br>000 |
| Only damages (n=1360)       | €1.00 | €119.<br>13 | €202.<br>57 | €498.<br>23 | €956.<br>75 | €1,72<br>7.25 | €3,470.<br>00 | €5,325.<br>44 | €17,703.4<br>4 | €169,305.<br>000 |

## **Table E2**. Detailed description insured rain damage data full dataset (from 2007).

| Variable                           | Mean<br>deviation if<br>parentheses | (standard<br>non-binary in | Median    |           | Range             |   |                  |
|------------------------------------|-------------------------------------|----------------------------|-----------|-----------|-------------------|---|------------------|
| From 2007                          | From 2007                           | From 2016                  | From 2007 | From 2016 | From 2007         |   | From 2016        |
| Insured rain damage                | €202.12<br>(€1928.92)               | €242.32<br>(€3029.46)      | €0.00     | €0.00     | €0<br>€169,305.00 | - | €0 – €169,305.00 |
| Insured rain damage (damages only) | €1867.82<br>(€5594.01)              | €2191.74<br>(€8883.30)     | €956.73   | €1000.00  | €1<br>€169,305.00 | _ | €1 – €169,305.00 |

Table E3. Distribution of rain data (from 2007).

|                                              | 1% | 5% | 10% | 25% | 50% | 75% | 90% | 95% | 99% | Largest |
|----------------------------------------------|----|----|-----|-----|-----|-----|-----|-----|-----|---------|
| Sum of rain per day (in 0.1 mm, n=12568)     | 0  | 0  | 0   | 0   | 1   | 26  | 76  | 115 | 202 | 672     |
| Maximum rain in an hour (in 0.1 mm, n=12568) | 0  | 0  | 0   | 0   | 1   | 11  | 26  | 39  | 89  | 281     |

## Appendix F. DiD regression with only significant variables.

**Table F1.** Two-way fixed effects DiD regression on insured damage per day in case of maximum rain per hour exceeds 2mm per hour from 2016 with only significant variables.

|                                 | (1)          |        | (2)           |        |
|---------------------------------|--------------|--------|---------------|--------|
| Variables                       | Model 1      | with   | Model 2       | with   |
|                                 | intervention | period | intervention  | period |
|                                 | (2007-2024)  |        | (2016 - 2024) |        |
| Post × treatment (DiD)          | -646.6*      |        | -4,188**      |        |
|                                 | (392.3)      |        | (1,626)       |        |
| Sum of rain per day (in 0.1 mm) | 5.562***     |        |               |        |
|                                 | (1.424)      |        |               |        |
| Population density              |              |        | -6.961**      |        |
|                                 |              |        | (2.798)       |        |
| Constant                        | -35.75       |        | 98,843**      |        |
|                                 | (196.0)      |        | (39,437)      |        |
| Observations                    | 1,766        |        | 886           |        |
| R-squared                       | 0.253        |        | 0.266         |        |

Standard errors in parentheses

510

<sup>\*\*\*</sup> p<0.01, \*\* p<0.05, \* p<0.1