# Peer review of "Assessing effects of nature-based and other municipal adaptation measures on insured heavy rain damages"

_EGUsphere, 2025_

## Referee Comment (RC1)

**Referee's Report: Assessing effects of nature-based and other municipal adaptation measures on insured heavy rain damages**

The study examines the effect of pluvial flood adaptation measures by analyzing insurance claims in two adjacent neighborhoods in the Netherlands before and after the interventions. The authors find that the adaptation measures in their totality cause a decline of 3700 euros per rainy day.

I agree with the authors that we still know too little about the effectiveness of mitigation measures, especially from an economic perspective. Studying the governmental mitigation projects adds to both the academic literature and has high social relevance. The use of insurance data provides a unique and objective measure to study the effectiveness of such measures.

While I think the set-up is interesting, multiple questions regarding the method and economic significance of the findings remain. Addressing these points should largely contribute to the paper, in my opinion. I recommend a revise and resubmit with major revisions.

**Methods**

I don't think that the visual interpretation around Figure 1 provides convincing evidence that the vital common pre-trend assumption is met. The study would largely benefit from a more formal test of the common pre-trend assumption. For example, through an event study.

Related to this, I wonder if it is likely that the amount of rainfall varies between your treatment and control group? If purely coincidental, the treatment group was affected by a large shower while the control group was not this biases results.

Both your treatment and control groups are affected by the general information campaign. If the effect of this campaign is homogenous for both areas, this should not affect the estimates that you get for the specific projects. However, if this is not the case, your results might be biased. In this regard, I think it is interesting to think about why the government chose to implement mitigation measures in the *Scheldebuurt* and not (yet) in the *Rijnbuurt*? This is ideally random. However, I can imagine that in this case, the firmer was chosen because the initial risk was higher, and thus the information campaign might also be more effective here.

Account for the insurance provider composition. The mitigation measures might be ex-ante observed by insurers who update their fees. Hence, the insurer composition before and after treatment might be altered. If certain insurers receive fewer claims in general (e.g., due to less coverage and user-friendliness of the claim portal), this could drive results.

Can spill-over effects bias the results? Many of the nature-based adaptation measures could also reduce risk in the control areas. For example, if the sewers are linked, increasing water storage capacity in the treatment area also reduces the chance that the water-bearing capacity in the control area is reached.

If I understand correctly, "post" is based on the date when the mitigation program started. However, realizing these projects does not happen instantly, and probably not all projects were implemented at the same time. Leaving out the intervention period should provide a more reliable estimate of the actual benefit, as the current measure likely underestimates the effect.

Time-specific neighborhood-level shocks are controlled for through fixed effects for each month. This addresses general temporal variation but not temporal variation at the neighborhood level. This would require 'month by neighborhood' FE.

Does the control percentage of real estate built before 1945 add new info? Unless a lot of new buildings were added during the study period, all this information should already be absorbed by the PC4 fixed effects. Also, it is unclear how the "address density" and the "value of property" controls are measured. Are those PC4 averages? If not, how can you use singular-level controls in combination with an aggregated dependent variable?

**Economic significance**

The authors state that their outcomes can be used to make a cost-benefit analysis. I would encourage them to do this already. If costs are difficult to obtain, at least provide some intuition on the benefits. What is the economic significance of saving 3696 euros per rainy day? Is that a lot or a little? How many rainy days are there?

Rather than controlling for rainfall, can you not incorporate this in your measure? E.g., a triple diff with post x treatment x sum of rain? Likely, the benefits from mitigation are not equal for different levels of rain. E.g., the measures only work to a certain limit, or they become even more effective beyond a certain amount of rain. It would be interesting to disentangle this.

The authors claim that studying multiple interventions in a single study is a strength. I would say that knowing about individual intervention contributions adds more value, as it allows for more optimal adaptation given budget constraints. Within the study set-up, it might be difficult to disentangle the benefits of individual adaptation measures. However, I would like to see some discussion on this. Do all measures equally contribute, are some more important than others, or is it exactly the joint effect that makes these measures so effective? Disentangling these added values would help with more efficient adaptation.

It is unclear whether the study utilizes all insurance claims or only those accepted. The latter is probably a cleaner measure. However, it can mask real damages and thus underestimate the net effect or bias results entirely if certain insurers reject more claims than others, and the composition of insurers changes during the study period. Related to this, I would like some reflection on how well insurance claims as a whole measure the effectiveness of mitigation measures. How large is the share of people not making claims despite damages, and what does this imply for your results?

---

## Referee Comment (RC2)

Review of

**Assessing effects of nature-based and other municipal adaptation**

**measures on insured heavy rain damages**

Author(s): Vylon Ooms, Thijs Endendijk, Jeroen. C.J.H. Aerts, W. J. Wouter Botzen, Peter Robinson

Manuscript No.: egusphere-2025-1882

Manuscript type: Research article

**General comments**

The manuscript evaluated the aggregated impact of a bundle of nature-based and other municipal adaptation measures on insured rain damages by comparing the insured data from two adjacent areas within the Rivierenbuurt neighbourhood, one with flood damage mitigation (FDM) measures and one without intervention. Using the statistical difference-in-difference method, the authors identified significant relations in some variables, highlighting the causal effect of FDM measures. The discussion and conclusions were valid and supported by data evidence.

The manuscript proposes a research initiative on a topic within the scope of Natural Hazards and Earth System Sciences (NHESS). I would recommend this manuscript for publication with the following suggestions, particularly regarding data processing, model interpretation, and widening the discussion.

**Specific comments**

1. Data heterogeneity and process
   - The listed adaptation measures in Table 1 may be applied to a specific area instead of the whole region. It is unclear how the local effects are translated to the larger treatment or local area. Moreover, given the spatial-temporal distribution of insured damage, is it possible to identify the relationship between adaptation measures and the observed reduced damages?
   - The study collected heterogeneous data (e.g., hydro-meteorological data and demographic characteristics) in space and time. It is unclear how authors address the heterogeneity and aggregate across the areas.

2. Data distribution
   - It is noted in Table 2, the standard deviation of insured rain damage was much higher than the mean. Is it driven by some extreme events that cause extensive insured rain damage? The same comment applies to "Rain data" in Table 3.
   - Related to Figure 1, the highest peak occurred in 2010. Was it due to any specific event or insurance claim?
   - Related to Tables 2 and 3, it will be nice, if possible, to visualise the data distribution through histograms.

3. Model development
   - Based on the results in Table 4, some variables which were conceived to be significant turned out to be not so significant (e.g., $p$-value $>0.1$) statistically in both models. Why the authors choose to keep these variables?
   - Some variables were correlated to some extent, e.g., three variables related to the sum of rain per day. Can this statistical modelling handle correlated variables? It will be nice to clarify the statistical assumptions.
   - There are other factors related to disaster management and capacity building (e.g., social vulnerability and infrastructure interruption) which is not considered in this study. If these factors were included, what would be the potential impact on the developed model and its implications?

4. Results
   - In the DiD model, the time and unit fixed effect is not well discussed. It would be nice to understand its implications.
   - Though the variables were discussed regarding their significance levels, the model performance (e.g., how well it fit the dependent variables) was not shown to visualise the goodness of fit and uncertainty. It is important to extend the discussion on the discussion, such as what other factors should be considered.
   - The placebo results in Appendix B it is hard to understand for a reader without a statistical background. A short description to guide readers through the results is necessary.
   - It is interesting to see the different trend patterns during the implementation of the adaptation measures. However, it is unclear whether it is due to the implementation temporarily reducing the overall adaptation effect or other reasons.

5. Discussion
   - This study evaluated the effectiveness of these measures combined in a municipality, only measured in terms of insured rain damage. However, it is worth providing insights regarding the long-term climate-adaptive benefit as well as the non-monetary impact.
   - A broad range of adaptation measures was studied as a whole. Can the data show the single contribution of respective measures to the overall climate-adaptation effect? Can the authors identify the most effective adaptation measures among all the considered measures?

---

## Author Comment (AC1)

**Response to Reviewer #1**

We thank the reviewer for the positive evaluation of the paper and the thorough review. Each comment is addressed separately and in detail below. The original comments of the reviewers are given in italics, followed by our response in normal letter type.

*Referee's Report: Assessing effects of nature-based and other municipal adaptation measures on insured heavy rain damages*

*The study examines the effect of pluvial flood adaptation measures by analyzing insurance claims in two adjacent neighborhoods in the Netherlands before and after the interventions. The authors find that the adaptation measures in their totality cause a decline of 3700 euros per rainy day.*

*I agree with the authors that we still know too little about the effectiveness of mitigation measures, especially from an economic perspective. Studying the governmental mitigation projects adds to both the academic literature and has high social relevance. The use of insurance data provides a unique and objective measure to study the effectiveness of such measures.*

*While I think the set-up is interesting, multiple questions regarding the method and economic significance of the findings remain. Addressing these points should largely contribute to the paper, in my opinion. I recommend a revise and resubmit with major revisions.*

**Methods**

*I don't think that the visual interpretation around Figure 1 provides convincing evidence that the vital common pre-trend assumption is met. The study would largely benefit from a more formal test of the common pre-trend assumption. For example, through an event study.*

We thank the reviewer for drawing our attention to this important point. Based on their comments, we plan to delete figure 1 from the main text (now in appendix 4). We would like to draw the reviewer's attention to the placebo test results, which is a formal test of whether the common trend assumption is met. The detailed results are shown in Appendix 2. The following sentences are part of the revised section 2.4:

'(...) a placebo test can be performed to check for the common trend assumption (Eggers et al., 2021). The placebo test checks the common trend assumption by creating "fake" treatment groups before and after the intervention. We select a different treatment timeframe and observe whether the effects are significant as well. If no effect is found in any of the placebo groups, it supports that the found treatment effect can be

attributed to the treatment rather than pre-existing trends. Angrist and Pischke (2008) used lag and lead values of treatment status to show that no significant effects occurred in the placebo periods. In Appendix 2, we apply placebo tests by using one- and two-month leads and lags for the treatment variable. These placebo treatment variables resulted in non-significant outcomes, reinforcing the validity of the common trend assumption for causal inference.'

*Related to this, I wonder if it is likely that the amount of rainfall varies between your treatment and control group? If purely coincidental, the treatment group was affected by a large shower while the control group was not this biases results.*

The rain data we used from the Royal Dutch Meteorological Office is the same for both the treatment and control group. In both cases the nearest weather station is used. We expect only very minor differences in actual rainfall between the treatment and control group, since they are neighborhoods adjacent to one another. Therefore, they likely receive similar rainfall volumes and extremes. Pre-existing infrastructure is also similar, where only in the treatment area measures were taken to reduce rain damage (Amsterdam Weerproof, 2024).

*Both your treatment and control groups are affected by the general information campaign. If the effect of this campaign is homogenous for both areas, this should not affect the estimates that you get for the specific projects. However, if this is not the case, your results might be biased. In this regard, I think it is interesting to think about why the government chose to implement mitigation measures in the Scheldebuurt and not (yet) in the Rijnbuurt? This is ideally random. However, I can imagine that in this case, the firmer was chosen because the initial risk was higher, and thus the information campaign might also be more effective here.*

We do not have detailed information about why the Scheldebuurt (treatment) was chosen first. According to the municipality, the sewage- and rainwater system needed to be renewed. In both neighborhoods, large areas were once covered with concrete and tiles, thus reducing water drainage (Amsterdam Weerproof, 2024). The municipality started implementing FDM measures in Rijnbuurt (control) from 2025 (Amsterdam Weerproof, 2024). The information campaigns are the same across both neighborhoods, (Amsterdam Weerproof, 2025). We thus expect the effect of information provision to have a similar effect in both neighborhoods. We plan to incorporate a sentence in the paper that the treatment effects are attributed to the other FDM measures only. The impact of advice through information campaigns was expected to be low based on previous research (Osberghaus & Hinrichs, 2021), compared to tangible community focused measures like constructing green areas and water buffering zones.

*Account for the insurance provider composition. The mitigation measures might be ex-ante observed by insurers who update their fees. Hence, the insurer composition before and after treatment might be altered. If certain insurers receive fewer claims in general (e.g., due to less coverage and user-friendliness of the claim portal), this could drive results.*

> In practice, Dutch insurers do not update their fees based on mitigation measures (Kroes & Klok, 2024). Even though an insurance market changes and develops over time, rain damages were already widely insurable since the adoption of the so called Neerslagclausule (precipitation clause) in 2001 (Dutch Association of Insurers, 2018). For households and businesses, rain damage is insured by default (Dutch Association of Insurers, 2025). The data of households of all members of the Dutch Association of Insurers (over 95% of the Dutch market) is used (Dutch Association of Insurers, 2024). Also, we do not expect the insurer composition to change much over time. In the Netherlands, households tend to not switch insurance policies often. There is no data available for households changing insurer for non-life insurers, but for health insurers this is about 7.4% per year (Zorgverzekeraars Nederland, 2025). It is expected that this is much lower for non-life insurances, since health insurers actively campaign every year in december to change insurer. Also, for car insurance, households tend to stay for more than 12,5 years on average at the same insurer (Allianz Direct, 2025). Therefore, we expect that insurer composition does not change much across time.

*Can spill-over effects bias the results? Many of the nature-based adaptation measures could also reduce risk in the control areas. For example, if the sewers are linked, increasing water storage capacity in the treatment area also reduces the chance that the water-bearing capacity in the control area is reached.*

> We would argue that a possible spill-over effect, would be very minor. The measures are primarily focused on infiltration (e.g. green areas, water storage, green roofs) on a very local level, specifically focused on the 'Scheldebuurt' (treatment area). Large scale measures such as like dikes or large water barriers could influence groundwater flows across larger areas. However, these measures are not taken in the treatment area. Furthermore, most damages (about 60%) are roof leakages from rain, not coming from ground water (Amsterdam Weerproof, 2014). A green roof or similar measures could help prevent these damages. For these damages, no spill-over effect to another area occurs.

*If I understand correctly, "post" is based on the date when the mitigation program started. However, realizing these projects does not happen instantly, and probably not all projects were implemented at the same time. Leaving out the intervention period*

*should provide a more reliable estimate of the actual benefit, as the current measure likely underestimates the effect.*

We agree with the reviewer that not including observations during the intervention period, would result in a cleaner analysis. We avoid potential bias from including the rollout period, when the policy's effect was only partial. Therefore, we ran the analysis without observations in the intervention period.

**Table 1: Two-way fixed effects DiD regression on insured damage per day in case of maximum rain per hour exceeds 2mm per hour from 2007 with and without observations in the intervention period**

| Variables | (1)
Model 1 (results with intervention period) | (2)
Model 2 (results without intervention period) |
|---|---|---|
| Post × treatment (DiD) | -647.0* | -1,375** |
| | (392.5) | (558.2) |
| Sum of rain per day (in 0.1 mm) | 6.267*** | 6.856*** |
| | (1.949) | (2.308) |
| Sum of rain per day lag 1 (in 0.1 mm) | -2.158 | -2.010 |
| | (3.004) | (3.635) |
| Sum of rain per day lag 2 (in 0.1 mm) | 0.434 | -0.0834 |
| | (3.423) | (4.274) |
| Maximum rain in an hour (in 0.1 mm) | -2.679 | -3.053 |
| | (5.075) | (6.315) |
| Maximum rain in an hour lag 1 (in 0.1 mm) | 10.62 | 10.88 |
| | (7.733) | (9.674) |
| Maximum rain in an hour lag 2 (in 0.1 mm) | -2.997 | -0.665 |
| | (9.572) | (12.79) |
| Constant | -61.99 | -61.37 |
| | (240.2) | (286.0) |
| | | |
| Observations | 1,766 | 1,416 |
| R-squared | 0.254 | 0.259 |
| Adjusted R-squared | 0.162 | 0.167 |

Standard errors in parentheses
*** $p<0.01$, ** $p<0.05$, * $p<0.1$

We performed multi-collinearity tests, which showed that the previous composition of variables resulted in multi-collinearity problems for the variables 'average number of people per household per address', 'percentage of real estate built before 1945' and 'address density' (all higher than 0.8). Therefore, we plan to remove the first two of these variables from the analysis and we plan to change 'address density' to a different variable called 'population density': the amount of people per area divided by the size of the area. The results are shown below. Model 1 shows the results with intervention period and model 2 without intervention period with the observations from 2007-2024. Model 3 (with intervention period) and model 4 (without intervention period) show the data from 2016-2024 with the extra variables 'population density' and 'value of property'.

**Table 2: Two-way fixed effects DiD regression on insured damage per day in case of maximum rain per hour exceeds 2mm per hour from 2016 with and without observations in the intervention period**

| Variables | (3) Model 3 (results from 2016 with intervention period) | (4) Model 4 (results from 2016 without intervention period) |
|---|---|---|
| Post × treatment (DiD) | -5,017*** | -5,648** |
| | (1,863) | (2,512) |
| Sum of rain per day (in 0.1 mm) | 5.405 | 7.100 |
| | (3.650) | (5.375) |
| Sum of rain per day lag 1 (in 0.1 mm) | -1.136 | -0.986 |
| | (5.774) | (9.271) |
| Sum of rain per day lag 2 (in 0.1 mm) | 2.537 | 0.624 |
| | (6.711) | (11.80) |
| Maximum rain in an hour (in 0.1 mm) | -8.205 | -13.90 |
| | (9.669) | (16.89) |
| Maximum rain in an hour lag 1 (in 0.1 mm) | 11.49 | 14.21 |
| | (15.41) | (27.48) |
| Maximum rain in an hour lag 2 (in 0.1 mm) | -8.074 | 2.483 |
| | (18.22) | (37.70) |
| Population density (per km²) | -7.325*** | -6.391 |
| | (2.827) | (5.845) |
| Value of property (in euros) | 24.67 | 48.00 |
| | (27.79) | (56.33) |
| Constant | 91,656** | 66,643 |
| | (40,384) | (98,878) |
| | | |
| Observations | 886 | 536 |
| R-squared | 0.269 | 0.271 |
| Adjusted R-squared | 0.174 | 0.173 |

Standard errors in parentheses
*** p<0.01, ** p<0.05, * p<0.1

The reviewer argues that the model without observations in the interaction period would result in a cleaner analysis. Also, the fit of one model improves compared to the analysis with observations in the intervention period (model 1). Therefore, we plan to use the models without observations in the intervention period for the analysis. The models with observations in the intervention period we plan to use as robustness tests. Therefore, we plan to show the following models without intervention period in the results chapter:

**Table 3: Two-way fixed effects DiD regression on insured damage per day in case of maximum rain per hour exceeds 2mm per hour from 2007 and 2016 without observations in the intervention period**

| Variables | (1) Model 1 | (2) Model 2 |
|---|---|---|
| Post × treatment (DiD) | -1,375** | -5,648** |
| | (558.2) | (2,512) |
| Sum of rain per day (in 0.1 mm) | 6.856*** | 7.100 |
| | (2.308) | (5.375) |

| | | |
|---|---|---|
| Sum of rain per day lag 1 (in 0.1 mm) | -2.010 | -0.986 |
| | (3.635) | (9.271) |
| Sum of rain per day lag 2 (in 0.1 mm) | -0.0834 | 0.624 |
| | (4.274) | (11.80) |
| Maximum rain in an hour (in 0.1 mm) | -3.053 | -13.90 |
| | (6.315) | (16.89) |
| Maximum rain in an hour lag 1 (in 0.1 mm) | 10.88 | 14.21 |
| | (9.674) | (27.48) |
| Maximum rain in an hour lag 2 (in 0.1 mm) | -0.665 | 2.483 |
| | (12.79) | (37.70) |
| Population density (per km²) | | -6.391 |
| | | (5.845) |
| Value of property (in euros) | | 48.00 |
| | | (56.33) |
| Constant | -61.37 | 66,643 |
| | (286.0) | (98,878) |
| | | |
| Observations | 1,416 | 536 |
| R-squared | 0.259 | 0.271 |
| Adjusted R-squared | 0.167 | 0.173 |

Standard errors in parentheses
*** p<0.01, ** p<0.05, * p<0.1

*Time-specific neighborhood-level shocks are controlled for through fixed effects for each month. This addresses general temporal variation but not temporal variation at the neighborhood level. This would require 'month by neighborhood' FE.*

We believe that time fixed effects per neighborhood may absorb or obscure some of the intervention effects. General time fixed effects control for common shocks or trends over time (this is why we opt to include these effects in our analysis). Neighborhood-by-time fixed effects are likely to control for time-specific changes within each neighborhood. Therefore, we plan to retain the models presented under the above comment.

*Does the control percentage of real estate built before 1945 add new info? Unless a lot of new buildings were added during the study period, all this information should already be absorbed by the PC4 fixed effects.*

See the previous comment about the models. We decided to remove the variable 'real estate built before 1945' due to multi-correlation problems with this variable.

*Also, it is unclear how the "address density" and the "value of property" controls are measured. Are those PC4 averages? If not, how can you use singular-level controls in combination with an aggregated dependent variable?*

See our answer related to multi-collinearity above. Specifically, the previous composition of variables resulted in multi-collinearity problems for the variables 'average number of people per household per address', 'percentage of real estate

built before 1945' and 'address density'. Therefore, we removed the first two variables from the analysis and changed 'address density' to a different variable called 'population density': the amount of people per area divided by the size of the area. The inclusion of these two control variables does not result in multi-collinearity.

We agree with the reviewer that the description of 'address density' and 'value of property' could be improved.

On the variable we plan to add 'Population density:
'The amount of people within each neighborhood per km$^2$.'

On 'Value of property':
'Average price per real estate asset per neighborhood based on the Valuation of Immovable Property Act (WOZ) (€x1000)."

**Economic significance**
*The authors state that their outcomes can be used to make a cost-benefit analysis. I would encourage them to do this already. If costs are difficult to obtain, at least provide some intuition on the benefits. What is the economic significance of saving 3696 euros per rainy day? Is that a lot or a little? How many rainy days are there?*

We agree with the reviewer that interpreting a result like 'saving an x-amount of euros per rainy day' can be difficult. Therefore, we plan to add more information that helps understanding the results more clearly. We plan to look at days that resulted in extreme damages and estimate the average total number of extreme rain days that may result in damage on an annual basis. In this way, the result can be expressed in damage prevented annually in the treatment area.

*Rather than controlling for rainfall, can you not incorporate this in your measure? E.g., a triple diff with post x treatment x sum of rain? Likely, the benefits from mitigation are not equal for different levels of rain. E.g., the measures only work to a certain limit, or they become even more effective beyond a certain amount of rain. It would be interesting to disentangle this.*

We tested for this by including a three-way interaction: post x treatment x sum of rain. The results are different from the previous results. This might be the case because ''sum of rain'' only takes the day into account on which damage was registered. However, in the models we deliberately use lags of one day and two days, to account for late registrations of damages that actually occurred one or two days earlier.

*The authors claim that studying multiple interventions in a single study is a strength. I would say that knowing about individual intervention contributions adds more value, as it allows for more optimal adaptation given budget constraints. Within the study set-up, it*

*might be difficult to disentangle the benefits of individual adaptation measures. However, I would like to see some discussion on this. Do all measures equally contribute, are some more important than others, or is it exactly the joint effect that makes these measures so effective? Disentangling these added values would help with more efficient adaptation.*

We agree with the reviewer that it would be interesting to look at individual measures. We point this out in the limitations section, where we state the following:

'Lastly, this study shows the impact of all adaptation measures combined. In a future study, it might be of value to understand the impact of these measures separately.'

There are papers that look into the effects of single measures, for instance an old stormwater system (Sörensen & Emilsson, 2019), blue-green roofs (Busker et al., 2021) or awareness campaigns (Osberghaus & Hinrichs, 2020). A unique characteristic of this paper is that we look at all the measures combined. Municipal adaptation measures tend to focus on a package of measures in reality, not on one measure only. Given the information we have about the measures from the municipality, we cannot research the impact of a measure individually. Therefore, we plan to add the following sentence to the limitations section:

'Further, it would be valuable to understand how much separate measures contribute to damage reduction. This would give information on which measures policymakers could prioritize.'

*It is unclear whether the study utilizes all insurance claims or only those accepted. The latter is probably a cleaner measure. However, it can mask real damages and thus underestimate the net effect or bias results entirely if certain insurers reject more claims than others, and the composition of insurers changes during the study period. Related to this, I would like some reflection on how well insurance claims as a whole measure the effectiveness of mitigation measures. How large is the share of people not making claims despite damages, and what does this imply for your results?*

We look at the claims that are registered by the insurer. We plan to add the following line to section 2.1.1:

' The Dutch Association of Insurers registers claims of households filed by insurance companies that are member of the association. Since rain damage is covered by default (Dutch Association of Insurers, 2025), we expect that the vast majority of the claims are accepted.'

We do not know the percentage of the people who are insured but do not claim their damages, since there are no public numbers available on this topic. We argue that most people claim their damage when they are insured, and that when they are insured they will get compensated. Rain coverage is by default part of property and contents insurance products in the Netherlands (Dutch Association of Insurers, 2025). Very minor damages (e.g. of a few euros) may not be claimed, but we expect these damages to not alter the main findings of our study. On the other hand, bad maintenance or negligence can be a ground to not accept a claim. Nevertheless, according to anecdotal evidence from the insurance industry, this does not occur often.

In the discussion, under section 4.3 'Limitations and research implications' we plan to add the following sentences:

'It would be of value to look into uninsured damages (e.g. public infrastructure) and claims of businesses as well. Insured damage of households is only a part of total damage of extreme rain, but can still give valuable insights into the effectiveness of FDM measures.'

**References**

Allianz Direct. Overstappen van autoverzekering: https://www.allianzdirect.nl/autoverzekering/overstappen-autoverzekering/, last access: 15 August 2025.

Amsterdam Weerproof. Rivierenbuurt buurtpagina: https://weerproof.nl/rivierenbuurt/, last access: 15 August 2025.

Amsterdam Weerproof. Bewonersbijeenkomst Rijnbuurt Oost: Bewonersbijeenkomst Rijnbuurt Oost - Weerproof, last access 15 August 2025.

Angrist, J. and Pischke, J.-S.: *Mostly Harmless Econometrics: An Empiricist's Companion*, Princeton University Press, Princeton, NJ, 392 pp., 2009.

Busker, T., de Moel, H., Haer, T., Schmeits, M., van den Hurk, B., Myers, K., Cirkel, D. G., and Aerts, J.: Blue-green roofs with forecast-based operation to reduce the impact of weather extremes, Journal of Environmental Management, 301, 1–12, https://doi.org/10.1016/j.jenvman.2021.113750, 2022.

Dutch Association of Insurers: Ledenlijst / Lid worden: https://www.verzekeraars.nl/over-het-verbond/lid-worden, last access: 15 August 2025.

Dutch Association of Insurers: Dutch Insurance Industry in Figures 2016: verzekerd-van-cijfers-2016-eng.pdf, last access 15 August 2025.

Dutch Association of Insurers, Overstroming en droogte: schade en verzekeringen: https://www.verzekeraars.nl/verzekeringsthemas/klimaatbestendig-verzekeren/overstroming-en-droogte, last access: 15 August 2025.

Eggers, A. C., Tuñón, G., and Dafoe, A.: Placebo Tests for Causal Inference, American Journal of Political Science, 68, 1106–1121, https://doi.org/10.1111/ajps.12818, 2024.

Kroes, S., Klok, L., and van de Kerkhof, A.: Enablers and Barriers of Nature-based Solutions, n.d.

Osberghaus, D. and Hinrichs, H.: The Effectiveness of a Large-Scale Flood Risk Awareness Campaign: Evidence from Two Panel Data Sets, Risk Analysis, 41, 944–957, https://doi.org/10.1111/risa.13601, 2021.

Skrydstrup, J., Löwe, R., Gregersen, I. B., Koetse, M., Aerts, J. C. J. H., de Ruiter, M., and Arnbjerg-Nielsen, K.: Assessing the recreational value of small-scale nature-based solutions when planning urban flood adaptation, Journal of Environmental Management, 320, 115724, https://doi.org/10.1016/j.jenvman.2022.115724, 2022.

Sörensen, J. and Emilsson, T.: Evaluating Flood Risk Reduction by Urban Blue-Green Infrastructure Using Insurance Data, Journal of Water Resources Planning and Management, 145, https://doi.org/10.1061/(ASCE)WR.1943-5452.0001037, 2019.

Zorgverzekeraars Nederland. Definitief overstapcijfer bekend: Definitief overstapcijfer bekend: 7,4% wisselt van zorgverzekeraar - Zorgverzekeraars Nederland, last access 15 August 2025.

---

## Author Comment (AC2)

**Response to Reviewer #2**

We thank the reviewer for the positive evaluation of the paper and the feedback. We address each comment separately and in detail below. The original comments of the reviewers are given in italics, followed by our response in normal letter type.

**General comments**

*General comments The manuscript evaluated the aggregated impact of a bundle of nature-based and other municipal adaptation measures on insured rain damages by comparing the insured data from two adjacent areas within the Rivierenbuurt neighbourhood, one with flood damage mitigation (FDM) measures and one without intervention. Using the statistical difference-in-difference method, the authors identified significant relations in some variables, highlighting the causal effect of FDM measures. The discussion and conclusions were valid and supported by data evidence.*

*The manuscript proposes a research initiative on a topic within the scope of Natural Hazards and Earth System Sciences (NHESS). I would recommend this manuscript for publication with the following suggestions, particularly regarding data processing, model interpretation, and widening the discussion.*

**Specific comments**

**Data heterogeneity and process**
*The listed adaptation measures in Table 1 may be applied to a specific area instead of the whole region. It is unclear how the local effects are translated to the larger treatment or local area. Moreover, given the spatial-temporal distribution of insured damage, is it possible to identify the relationship between adaptation measures and the observed reduced damages?*

We account for this relationship by selecting the Difference-in-Difference method. This method controls time-invariant unobserved differences between neighborhoods (in this case the treatment and control neighborhood), such as historical infrastructure and socioeconomic factors, as well as time-specific shocks, like extreme weather events.

Furthermore, due to privacy restrictions we are not allowed to look at the damages at the address level. It is therefore difficult to estimate the local effect of a measure, more detailed than neighborhood level (PC4). We plan to add a footnote in section 2.2.1.

'Due to privacy restrictions on the claims data it is not possible to analyze the damages on address level.'

*The study collected heterogeneous data (e.g., hydro-meteorological data and demographic characteristics) in space and time. It is unclear how authors address the heterogeneity and aggregate across the areas.*

> See response directly above. We discuss the characteristics of both neighborhoods in appendix 1.

> In the data description section (table 2 and 3) we describe for each variable (1) the time period between which the data is recorded; and (2) the level of aggregation, e.g. average at postcode or neighbourhood level.

***Data distribution***
*It is noted in Table 2, the standard deviation of insured rain damage was much higher than the mean. Is it driven by some extreme events that cause extensive insured rain damage? The same comment applies to "Rain data" in Table 3.*

> This is indeed driven by extreme events. Specifically, August 2010 was a month where extreme damages occurred. We performed a robustness test by deleting this month and rerunning the analysis again. However, this caused only minor changes to the results. We plan to add a footnote to describe this in section 4.1:

> > 'In an additional analysis, we omitted the month August 2010, where large damages occurred in the treatment group and the control group. We observe minor changes to the results: the interaction coefficient is -704.461, compared to the -646.963 in the model with August 2010 included, and the relation is significant at the same level ($p < 0.01$).'

> We also plan to add a sentence in section 4.1 linking the additional analysis, with the omission of August 2010, to the high standard deviation compared to the average damages (table 2) and rain data (table 3). We plan to add the following sentence:

> > 'This also explains why the standard deviation is very high compared to the average of damage data (table 2) and rain data (table 3).'

*Related to Figure 1, the highest peak occurred in 2010. Was it due to any specific event or insurance claim?*

> This was due to the damages in the month of August 2010. See response to the previous comment on how we addressed this.

*Related to Tables 2 and 3, it will be nice, if possible, to visualise the data distribution through histograms.*

We agree with the reviewer that more detailed information would be an addition to the paper. However, since it is damage data that is guided by extremes, the dataset contains large outliers. Distributions shown via histograms are then not too helpful, because these are far apart and low damage observations are highly clustered. We instead plan to show how the data is distributed via tables showing the damage percentiles. We plan to include the following tables in the paper in appendix 4:

**Table 1: Distribution insured rain damage data full dataset (from 2007)**

| | 1% | 5% | 10% | 25% | 50% | 75% | 90% | 95% | 99% | Largest |
|---|---|---|---|---|---|---|---|---|---|---|
| All observations (n=12568) | €0.00 | €0.00 | €0.00 | €0.00 | €0.00 | €0.00 | €169.00 | €1000.00 | €3761.00 | €169305.00 |
| Only damages (n=1360) | €1.00 | €119.13 | €202.57 | €498.23 | €956.75 | €1727.25 | €3470.00 | €5325.44 | €17703.44 | €169305.00 |

**Table 2**: **Detailed description insured rain damage data full dataset (from 2007)**

| Variable | Mean (standard deviation if non-binary in parentheses) | | Median | | Range | |
|---|---|---|---|---|---|---|
| *From 2007* | *From 2007* | *From 2016* | *From 2007* | *From 2016* | *From 2007* | *From 2016* |
| Insured rain damage | €202.12 (€1928.92) | €242.32 (€3029.46) | €0.00 | €0 .00 | €0 – €169305.00 | €0 – €169305.00 |
| Insured rain damage (damages only) | €1867.82 (€5594.01) | €2191.74 (€8883.30) | €956.73 | €1000.00 | €1 – €169305.00 | €1 – €169305.00 |

**Table 3**: **Distribution of rain data (from 2007)**

| | 1% | 5% | 10% | 25% | 50% | 75% | 90% | 95% | 99% | Largest |
|---|---|---|---|---|---|---|---|---|---|---|
| Sum of rain per day (in 0.1 mm, n=12568) | 0 | 0 | 0 | 0 | 1 | 26 | 76 | 115 | 202 | 672 |
| Maximum rain in an hour (in 0.1 mm, n=12568) | 0 | 0 | 0 | 0 | 1 | 11 | 26 | 39 | 89 | 281 |

***Model development***

*Based on the results in Table 4, some variables which were conceived to be significant turned out to be not so significant (e.g., p-value >0.1) statistically in both models. Why the authors choose to keep these variables?*

We believe it is important to hold certain control variables constant when analyzing the intervention effect. We describe in section 2.2.2 the addition of the lag variables to account for the fact that the claims data consists of observations where the claim was filed by the insurer. This is often the same day, but can also be one or two days later. We account for this by adding the lags of one and two days in the analysis of the variables 'maximum rain in an hour' and 'um of rain per day of one and two days in the analysis'.

If we only include the significant variables in model 1, we obtain the results below. We plan to add this result as a robustness test to appendix 5.

**Table 4: Two-way fixed effects DiD regression on insured damage per day in case of maximum rain per hour exceeds 2mm per hour from 2016 with only significant variables**

| Variables | (1) Model 1 with intervention period (2007-2024) | (2) Model 2 with intervention period (2016 – 2024) |
|---|---|---|
| Post × treatment (DiD) | -646.6* | -4,188** |
| | (392.3) | (1,626) |
| Sum of rain per day (in 0.1 mm) | 5.562*** | |
| | (1.424) | |
| Population density | | -6.961** |
| | | (2.798) |
| Constant | -35.75 | 98,843** |
| | (196.0) | (39,437) |
| Observations | 1,766 | 886 |
| R-squared | 0.253 | 0.266 |

Standard errors in parentheses
*** p<0.01, ** p<0.05, * p<0.1

*Some variables were correlated to some extent, e.g., three variables related to the sum of rain per day. Can this statistical modelling handle correlated variables? It will be nice to clarify the statistical assumptions.*

We tested for multicollinearity through the use of the Variance Inflation Factor (VIF) method. VIF is commonly used in econometric analyses (Woolridge, 2016). If a VIF-value is between 1 and 5, it is deemed as acceptable collinearity. If a VIF-value is above 10, we can conclude that multicollinearity is problematic for that variable (Woolridge, 2016). We performed a VIF-test for the models in the manuscript: model 1 and model 2. For model 1 we observe no issues (VIF-scores below 5). However, for model 2 we observe high VIF-values (>10) for the variables 'percentage of real estate built before 1945', 'Address density' and 'Average number of people per household per address'. We therefore plan to delete these variables from the previous model 2. The first reviewer commented that it would be cleaner to do the analysis with omitting the intervention period. We plan to follow this comment. The results of model 2, which we include in two models, with and without the intervention period, are now as follows:

**Table 5: Two-way fixed effects DiD regression on insured damage per day in case of maximum rain per hour exceeds 2mm per hour from 2016 with and without observations in the intervention period**

| Variables | (3) Model 3 (results from 2016 with intervention period) | (4) Model 4 (results from 2016 without intervention period) |
|---|---|---|
| Post × treatment (DiD) | -5,017*** | -5,648** |
| | (1,863) | (2,512) |
| Sum of rain per day (in 0.1 mm) | 5.405 | 7.100 |
| | (3.650) | (5.375) |
| Sum of rain per day lag 1 (in 0.1 mm) | -1.136 | -0.986 |
| | (5.774) | (9.271) |
| Sum of rain per day lag 2 (in 0.1 mm) | 2.537 | 0.624 |
| | (6.711) | (11.80) |
| Maximum rain in an hour (in 0.1 mm) | -8.205 | -13.90 |
| | (9.669) | (16.89) |
| Maximum rain in an hour lag 1 (in 0.1 mm) | 11.49 | 14.21 |
| | (15.41) | (27.48) |
| Maximum rain in an hour lag 2 (in 0.1 mm) | -8.074 | 2.483 |
| | (18.22) | (37.70) |
| Population density (per km²) | -7.325*** | -6.391 |
| | (2.827) | (5.845) |
| Value of property (in euros) | 24.67 | 48.00 |
| | (27.79) | (56.33) |
| Constant | 91,656** | 66,643 |
| | (40,384) | (98,878) |
| | | |
| Observations | 886 | 536 |
| R-squared | 0.269 | 0.271 |
| Adjusted R-squared | 0.174 | 0.173 |

Standard errors in parentheses
*** p<0.01, ** p<0.05, * p<0.1

With changing the variables, we also changed the variable description table of the control variables. We plan to add the description below.

**Table 6**: **Added control variable**

| Variable | Variable description | Data source | Mean and standard deviation if non-binary |
|---|---|---|---|
| *Area characteristics (per day from 2016)* | | | |
| Population density | The amount of people per km² | CBS | 13897.77 (744.84) |

*There are other factors related to disaster management and capacity building (e.g., social vulnerability and infrastructure interruption) which is not considered in this study.*

*If these factors were included, what would be the potential impact on the developed model and its implications?*

We understand the comment of the reviewer. To examine social vulnerability, one would probably need to collect and couple survey data on individual socio-economic characteristics and combine it with damage and rainfall data. This is practically difficult, given the level of aggregation of the damage (pc4) and rainfall data. For a future study, it could be interesting to understand social vulnerability as well. Given that social vulnerability might influence insurance uptake. In the discussion, under section 4.3 'Limitations and research implications' we plan to add the following sentence:

'Additionally, it would be insightful to look at social vulnerability, since that could influence insurance uptake.'

However, this should not impact the results, since external developments are the same in both neighborhoods. Socio-economic characteristics are likely similar across both neighborhoods. The DiD two-way fixed effects approach controls for socioeconomic differences. We test for the common trend assumption using a placebo test. We show the results of these tests in appendix 2. We explain the placebo tests in section 2.4. We see no influence of the factors mentioned by the reviewer, because the treatment and control groups follow a similar trend before the interventions (see placebo tests in appendix 2).

**Results**
*In the DiD model, the time and unit fixed effect is not well discussed. It would be nice to understand its implications.*

We agree with the reviewer that we could expand on the time and unit fixed effects explanation. The main implication is that fixed effects in a DiD are used to give a more robust causal estimate, controlling for time-invariant unobserved differences like socioeconomic factors (on unit) or extreme weather events (on time). We plan to alter the text in the manuscript as follows:

'In this study, we use a DiD two-way fixed effects model to estimate the impact of municipal adaptation measures on rainfall damage in Amsterdam. One can compare a situation before and after an intervention. We compare two adjacent areas within the Rivierenbuurt neighborhood: one where flood damage mitigation (FDM) measures have been implemented (Scheldebuurt) and another where no interventions have been implemented (Rijnbuurt). The DiD approach allows us to compare changes in outcomes over time between these areas, while controlling for unobserved factors and broader trends (Card & Krueger, 1993; Wooldridge, 2014). By leveraging insurance claims data, we can isolate the causal impact of these measures under the assumption that both areas would have followed

similar trends in the absence of interventions. We test this assumption in the next section.

We expand upon a traditional DiD by employing a two-way fixed effects (TWFE) model (Callaway & Sant'Anna, 2021). Using fixed effects in a DiD gives a more robust causal estimate. This approach controls for time-invariant unobserved differences between neighborhoods, such as historical infrastructure and socioeconomic factors, as well as time-specific shocks, like extreme weather events. By accounting for both unit (neighborhood) and time (month) fixed effects, the TWFE model ensures that our estimated treatment effect reflects the impact of adaptation measures rather than underlying trends or external influences. This strengthens the causal interpretation of the DiD analysis.'

DiD with time and unit fixed effects is an econometric method that is used to estimate causal effects. You compare a situation before and after a policy intervention. Fixed effects help to control for unobserved time-invariant heterogeneity. Unit fixed effects control for time-invariant characteristics of a unit, for instance geography of a location or individual differences.

Time fixed effects control for shocks that equally affect the units. For instance a recession, policy changes, or, in this case, extreme weather events for the whole group. Using fixed effects in a DiD gives a more robust causal estimate.

*Though the variables were discussed regarding their significance levels, the model performance (e.g., how well it fit the dependent variables) was not shown to visualise the goodness of fit and uncertainty.*

We show the Adjusted $R^2$ in table 4. We highlight this in the text by the following sentence in chapter 3, which we plan to change with the updated results:

'According to the adjusted $R^2$, Model 1 explains 16.7% of the variation in insured damage and model 2 explains 17.3% of the variation.'

*It is important to extend the discussion on the discussion, such as what other factors should be considered.*

We agree with the reviewer that this could be done more extensively. In the discussion, under section 4.3 'Limitations and research implications' we plan to add the following sentences:

'It would be of value to analyze uninsured damages and claims of businesses as well, to present a more complete picture of the effectiveness of FDM.'

'Furthermore, it would be valuable to understand how much separate measures contribute to damage reduction.'

Also, related to social vulnerability, we plan to add the following:

'Additionally, it would be insightful to look at social vulnerability, since that could influence insurance uptake.'

*The placebo results in Appendix B it is hard to understand for a reader without a statistical background. A short description to guide readers through the results is necessary.*

We agree it is useful to clarify these test results. We plan to add the following sentences to appendix 2 to guide the reader through the results shown in the tables of appendix 2:

The goal of this placebo test is to identify whether the treatment and control groups were experiencing similar trends before the treatment. This can be done by creating 'fake' treatments that indicate treatment before it actually occurred (Angrist & Pischke, 2009). These placebo treatments should have no effect if the common trend assumption holds. If they do show significant effects, this suggests a violation of the assumption, as it indicates that treated and control groups were already on diverging paths prior to the intervention.

We apply placebo tests by using one- and two-month leads and lags for the treatment variable. The lead and lagged placebo treatments do not show any significant outcomes, which provides evidence in favour of the common trend assumption.

*It is interesting to see the different trend patterns during the implementation of the adaptation measures. However, it is unclear whether it is due to the implementation temporarily reducing the overall adaptation effect or other reasons.*

We account for this in our method. The difference-in-difference (DiD) approach controls for time-invariant unobserved differences between neighborhoods, such as historical infrastructure and socioeconomic factors, as well as time-specific shocks, like extreme weather events. Homogenous shocks are experienced in both neighborhoods. The use of the DID approach makes it possible to only estimate the effect of the intervention.

**Discussion**
*This study evaluated the effectiveness of these measures combined in a municipality, only measured in terms of insured rain damage. However, it is worth providing insights regarding the long-term climate-adaptive benefit as well as the non-monetary impact.*

We agree with the reviewer that there are other benefits of these FDM measures next to rain damage reduction. Measures like water storage can be used against drought, green roofs and greener areas can mitigate heat and these areas can also be used for recreational purposes. These measures can limit long-term

impacts of climate change. We touch upon this in section 4.2 by the following sentences:

'Local governments can use nature based and other adaptation measures (e.g. through green lanes, water storage facilities, green roofs, and greener gardens) as means to decrease rain damage in urban areas and increase livability and biodiversity in these areas (Skrydstrup et al., 2022). These nature based measures often come with co-benefits like mental and physical benefits (Tzoulas et al., 2007).'

However, an example of a long term benefit is less taken into consideration. Therefore, we plan to add the following sentence:

'(... ), which can have a long term impact on health as well by incentivizing people to exercise for instance.'

In the current text we do not yet point out that the municipal climate adaptation measures not only limit rain damage, but also limit impacts of other natural hazards like drought and heat. We plan to add the following sentence to section 4.2:

'Rain damage is the focus of this study. The measures the municipality applied can also limit impacts of other natural hazards, like drought and heat. These measures can limit long-term impacts of climate change in the area.'

*A broad range of adaptation measures was studied as a whole. Can the data show the single contribution of respective measures to the overall climate-adaptation effect? Can the authors identify the most effective adaptation measures among all the considered measures?*

The reviewer is correct that it would be interesting to look at individual measures. We now point this out in the limitations section, where we plan to add the following text:

'Lastly, this study shows the impact of all adaptation measures combined. Because of privacy regulations, it was not possible to localize claims on a more detailed level than PC4-level. This makes it difficult to attach effects of local measures to single damage claims. In a future study, it might be of value to understand the impact of these measures separately.'

There are papers that assesses the risk reduction of a single measure, for instance an old stormwater system (Sörensen & Emilsson, 2019), blue-green roofs (Busker et al., 2021) or awareness campaigns (Osberghaus & Hinrichs, 2020). A unique characteristic of this paper is that we look at all the measures combined. Given how the measures were implemented, we cannot research the

impact of a measure individually. Therefore, we plan to add the following sentence to the limitations section:

'Further, it would be valuable to understand how much separate measures contribute to damage reduction.'

**References**

Busker, T., de Moel, H., Haer, T., Schmeits, M., van den Hurk, B., Myers, K., Cirkel, D. G., and Aerts, J.: Blue-green roofs with forecast-based operation to reduce the impact of weather extremes, Journal of Environmental Management, 301, 1–12, https://doi.org/10.1016/j.jenvman.2021.113750, 2022.

Angrist, J. and Pischke, J.-S.: *Mostly Harmless Econometrics: An Empiricist's Companion*, Princeton University Press, Princeton, NJ, 392 pp., 2009.

Callaway, B. and Sant'Anna, P. H. C.: Difference-in-Differences with multiple time periods, Journal of Econometrics, 225, 200–230, https://doi.org/10.1016/j.jeconom.2020.12.001, 2021.

Card, D. and Krueger, A. B.: Minimum Wages and Employment: A Case Study of the Fast-Food Industry in New Jersey and Pennsylvania: Reply, The American Economic Review, 90, 1397–1420, 2000.

Osberghaus, D. and Hinrichs, H.: The Effectiveness of a Large-Scale Flood Risk Awareness Campaign: Evidence from Two Panel Data Sets, Risk Analysis, 41, 944–957, https://doi.org/10.1111/risa.13601, 2021.

Sörensen, J. and Emilsson, T.: Evaluating Flood Risk Reduction by Urban Blue-Green Infrastructure Using Insurance Data, Journal of Water Resources Planning and Management, 145, https://doi.org/10.1061/(ASCE)WR.1943-5452.0001037, 2019.

Skrydstrup, J., Löwe, R., Gregersen, I. B., Koetse, M., Aerts, J. C. J. H., de Ruiter, M., and Arnbjerg-Nielsen, K.: Assessing the recreational value of small-scale nature-based solutions when planning urban flood adaptation, Journal of Environmental Management, 320, 115724, https://doi.org/10.1016/j.jenvman.2022.115724, 2022.

Tzoulas, K., Korpela, K., Venn, S., Yli-Pelkonen, V., Kaźmierczak, A., Niemela, J., and James, P.: Promoting ecosystem and human health in urban areas using Green Infrastructure: A literature review, Landscape and Urban Planning, 81, 167–178, https://doi.org/10.1016/j.landurbplan.2007.02.001, 2007.

Wooldridge, J. M.: Introduction to Econometrics (Europe, Middle East & Africa Edition), Cengage Learning EMEA, Andover, United Kingdom, 912 pp., 2014.

---

## Author Comment (AC3)

**Response to Reviewer #3**

We thank the reviewer for the positive comments and review. Each comment is addressed in detail directly under each comment of the reviewer below. The original comments of the reviewers are given in italics, followed by our response in normal letter type.

*Many quantitative ex-ante studies have indicated the value of NBS interventions in urban areas, because of various Ecosystem Services, including mitigatin of urban heat island impacts, reduced impacts on the water cycle and mitigating flood risk. Especially for mitigation of flood risk many studies have indicated detailed approaches for quantification of the expected impacts.*

*On the other hand very few studies have been perfomed ex-post to verify the validity of the assumptions of the investments made to install NBS interventions. This study is therefore highly welcomed because the scope is exactly what the flood risk modelling community has repeatedly requested: to perform ex-post analyses to enable structured learning and inform future projects of the learnings of past projects.*

*However, this is also the most important limitation of the study. I would therefore encourage the authors to expand and improve their study prior to finalizing the paper. Most importantly to include information from the ex-ante study leading to the NBS interventions: What were the expected reduction in Expected Annual Damage (EAD), how was this linked to properties of rainfall, and did the reductions in damages occur at the expected locations?*

We observe a reduction in the treatment area. Thus, the reduction occurred at the expected location. We agree with the reviewer that including an EAD would be of value. This would ask for a modelling study, which would be a new study that is outside the scope of this research.

To our knowledge, there are no publicly available sources on EAD and expectations of reduction of damage by the municipality. We therefore reached out to our contacts at the municipality of Amsterdam to ask if they could provide us with damage reduction expectations before the intervention. However, they replied that they could not provide us with this information. They did mention that the treatment and control areas are known for often having water nuisance after rain. We know the ex-ante research was more qualitative than a full technical assessment. Even though it would be interesting to write about expected damage reduction beforehand, we are not able to add this to the paper.

*Validating and/or falsifying assumptions of the ex-ante study would be a truly interesting study and probably the information is readily at hand from the documents leading to the*

*strategy of NBS implementation. The transferability would also be improved if changed to an indicator related to EAD.*

*Another limitation of the study is the use of insurance data for buildings. There are always losses that are not covered by insurance and hence this should be mentioned in the discussion of the findings.*

> We agree with the reviewer that this could be done more extensively.

> Most rain damage losses are covered by insurance. For households and businesses, rain damage is insured by default (Dutch Association of Insurers, 2025).  In our dataset, the data of households of all members of the Dutch Association of Insurers (over 95% of the Dutch market) is used (Dutch Association of Insurers, 2024). We plan to add the following line:

> ' The Dutch Association of Insurers registers claims of households filed by insurance companies that are member of the association. Since rain damage is covered by default (Dutch Association of Insurers, 2025), we expect that the vast majority of the claims gets accepted.'

> We do not know the percentage of the people who are insured but do not claim their damages, since there are no public numbers available on this topic. We argue that most people claim their damage when they are insured, and that when they are insured they will get compensated. Rain coverage is by default part of property and contents insurance products in the Netherlands (Dutch Association of Insurers, 2025). Very minor damages (e.g. of a few euros) may not be claimed, but we expect these damages to not alter the main findings of our study. On the other side, bad maintenance or negligence can be a ground to not accept a claim.

> In the discussion, under section 4.3 'Limitations and research implications' we plan to add the following sentences:

> 'It would be of value to look into uninsured damages (e.g.  public infrastructure) and claims of businesses as well. Insured damage of households is only a part of total damage of extreme rain, but can still give valuable insights into the effectiveness of FDM measures.'

> 'Further, it would be valuable to understand how much separate measures contribute to damage reduction.'

*Figure 1 is difficult to interpret. I assume it is the median of the daily damages for days with more than X mm rainfall for each year? Make it more explicit and use more space to explore this data, e.g. by plotting the distributions for a suitable year and also aggregating to EAD for each catchment by year.*

We agree with the reviewer that figure 1 is difficult to interpret. It was shown to add a visual representation to the placebo test. However, we plan to move the figure to the appendix and add tables on data description that are shown further below.

We do not have information about an EAD, and this information could not be provided by Amsterdam Weerproof. The initiative Amsterdam Weerproof started after the extreme precipitation in 2014 (then, called Amsterdam Rainproof).

*Based on the figure alone I would assume that there would be no impacts of the NBS. This points to a more general issue with providing the readers with enough information about the data to validate the findings. More plots of both input and output data and model residuals would be an asset.*

We agree with the reviewer that more information could be given about the data. However, since it is damage data that is guided by extremes, real life outliers can occur. Distributions are then not too helpful to interpret, because these are far apart. What can give more information are extensive tables with descriptive data. We expanded table 2 and table 3 with the following tables, that we plan to add to appendix 4:

**Table 1: Distribution insured rain damage data full dataset (from 2007)**

|  | 1% | 5% | 10% | 25% | 50% | 75% | 90% | 95% | 99% | Largest |
|---|---|---|---|---|---|---|---|---|---|---|
| All observations (n=12568) | €0.00 | €0.00 | €0.00 | €0.00 | €0.00 | €0.00 | €169.00 | €1000.00 | €3761.00 | €169305.00 |
| Only damages (n=1360) | €1.00 | €119.13 | €202.57 | €498.23 | €956.75 | €1727.25 | €3470.00 | €5325.44 | €17703.44 | €169305.00 |

**Table 2**: **Detailed description insured rain damage data full dataset (from 2007)**

| Variable | Mean (standard deviation if non-binary in parentheses) | | Median | | Range | |
|---|---|---|---|---|---|---|
| *From 2007* | *From 2007* | *From 2016* | *From 2007* | *From 2016* | *From 2007* | *From 2016* |
| Insured rain damage | €202.12 (€1928.92) | €242.32 (€3029.46) | €0.00 | €0 .00 | €0 – €169305.00 | €0 – €169305.00 |
| Insured rain damage (damages only) | €1867.82 (€5594.01) | €2191.74 (€8883.30) | €956.73 | €1000.00 | €1 – €169305.00 | €1 – €169305.00 |

**Table 3**: **Distribution of rain data (from 2007)**

|  | 1% | 5% | 10% | 25% | 50% | 75% | 90% | 95% | 99% | Largest |
|---|---|---|---|---|---|---|---|---|---|---|
| Sum of rain per day (in | 0 | 0 | 0 | 0 | 1 | 26 | 76 | 115 | 202 | 672 |

| 0.1 mm, n=12568) | | | | | | | | | | |
|---|---|---|---|---|---|---|---|---|---|---|
| Maximum rain in an hour (in 0.1 mm, n=12568) | 0 | 0 | 0 | 0 | 1 | 11 | 26 | 39 | 89 | 281 |

*Minor points*

*The interventions occur in 2018 in the text but in the figure it looks like the intervention is in 2017? Is the x-asis correct?*

> The line is not precisely on the year 2018, but slightly before. We adjusted the line of the intervention slightly, in order that it lines up with the year 2018 (even though the intervention started in November 2017). See the change below, where the grey line corresponds to the year 2018. We plan to move the graph from the main text in the manuscript to the appendix.

**Figure 1: Median of rain damage per year**

[Figure]

*You could consider removing 2010 from the study all together, it would make for a more robust analysis.*

> The reviewer is correct. This is indeed driven by extreme events. Specifically, August 2010 was a month where extreme damages occurred. We performed a robustness test by deleting this month and rerunning the analysis again. However, this caused only minor changes to the results. We plan to add a footnote to describe this in section 4.1:

> ' In an additional analysis, we omitted the month August 2010, with the large damages in the control group. This month is an outlier and seemed to impact the interaction result and the coefficient. We observe minor changes in the results:

the interaction coefficient is -704,461, compared to the -646,963 in the model with August 2010 included, and the relation is significant on the same level ($p < 0.01$).'

**References**

Dutch Association of Insurers, Overstroming en droogte: schade en verzekeringen: https://www.verzekeraars.nl/verzekeringsthemas/klimaatbestendig-verzekeren/overstroming-en-droogte, last access: 1 August 2024.

Dutch Association of Insurers: Ledenlijst / Lid worden: https://www.verzekeraars.nl/over-het-verbond/lid-worden, last access: 18 March 2025.

Dutch Association of Insurers: Dutch Insurance Industry in Figures 2016: verzekerd-van-cijfers-2016-eng.pdf, last access 18 March 2025.